# Efficient optical plasmonic tweezer-controlled single-molecule SERS characterization of pH-dependent amylin species in aqueous milieus

Wenhao Fu[1,4], Huanyu Chi[1,4], Xin Dai[1,2], Hongni Zhu[1], Vince St. Dollente Mesias [1], Wei Liu [3] ✉ & Jinqing Huang [1] ✉

It is challenging to characterize single or a few biomolecules in physiological milieus without excluding the influences of surrounding environment. Here we utilize optical plasmonic trapping to construct a dynamic nanocavity, which reduces the diffraction-limited detection volume and provides reproducible electromagnetic field enhancements to achieve high-throughput single-molecule surface-enhanced Raman spectroscopy (SERS) characterizations in aqueous environments. Specifically, we study human Islet Amyloid Polypeptide (amylin, hIAPP) under different physiological pH conditions by combining spectroscopic experiments and molecular dynamics (MD) simulations. Based on a statistically significant amount of time-dependent SERS spectra, two types of low-populated transient species of hIAPP containing either turn or β-sheet structure among its predominant helix-coil monomers are characterized during the early-stage incubation at neutral condition, which play a crucial role in driving irreversible amyloid fibril developments even after a subsequent adjustment of pH to continue the prolonged incubation at acidic condition. Our results might provide profound mechanistic insight into the pH-regulated amyloidogenesis and introduce an alternative approach for investigating complex biological processes at the single-molecule level.

By removing ensemble averaging, single-molecule techniques discern the signal of individual molecules to unveil hidden details and revolutionize our understandings of physics, chemistry, and biology[1–3]. Of particular interest is the characterization of single or a few biomolecules, such as intrinsically disordered proteins (IDPs), in an aqueous milieu containing hydrogen-bonding, electrostatic, and hydrophilic-hydrophobic interactions during complex biological processes. For example, human Islet Amyloid Polypeptide (amylin, hIAPP) lacks stable secondary structures but possesses the aggregation propensity governed by its intrinsic sequence and surrounding environment to form amyloid fibrils with distinct β-sheet structures in type II diabetes patients[4]. The aggregation of hIAPP is repressed in the secretory granules of pancreatic β-cells at pH 5.5 and millimolar (mM) concentration while promoted in certain extracellular compartments at pH 7.4 and micromolar (μM) concentration[5], generating various oligomeric intermediates at low populations in a dynamic mixture towards fibril formation[6]. The crucial role of hIAPP oligomers in causing cytotoxicity and driving amyloid aggregation is widely

[1]Department of Chemistry, The Hong Kong University of Science and Technology, Clear Water Bay, Hong Kong, China. [2]Laboratory for Synthetic Chemistry and Chemical Biology, Health@InnoHK, Hong Kong Science Park, Hong Kong, China. [3]State Key Laboratory of Synthetic Chemistry, Department of Chemistry, The University of Hong Kong, Pokfulam Road, Hong Kong, China. [4]These authors contributed equally: Wenhao Fu, Huanyu Chi. ✉e-mail: wliu276@hku.hk; jqhuang@ust.hk

recognized in literatures[4,6,7]. However, owing to their transient nature and structural heterogeneity, it is challenging to characterize the detailed structures of hIAPP oligomers and the conversion from monomers to oligomers in complex amyloid assembly processes without excluding the influences of surrounding environment[4,8]. Although significant advancements have been achieved by single-molecule fluorescence methods, the structural determination is limited by fluorophore labeling, and the single-molecule scheme is restricted to ultra-dilution and/or molecular immobilization because the diffraction-limited detection volume cannot be further reduced[9].

Upon excitation of the localized surface plasmonic resonance (LSPR) in metallic nanostructures, surface-enhanced Raman spectroscopy (SERS) can overcome the optical diffraction-limit and recognize endogenous molecular vibrations[10] for chemical bond imaging[11], catalytic reaction monitoring[12], and DNA sequencing at the single-molecule level[13]. Since water generates weak Raman signals as background, it is feasible to operate in aqueous conditions. Nevertheless, the applications of single-molecule SERS are mainly reported at ultralow concentrations (aM-nM) and/or conducted on dry states without solvents[14-16], while the characterization of a single biomolecule involving in molecular interactions from aqueous environments at relatively high concentrations (µM-mM for physiological protein assembly and enzyme activity) remains challenging[9], considering the following issues: (i) Poor spatial control of the SERS active-detection volume[10]. (ii) Low stability of the plasmonic substrates suspended in liquid (e.g., nanoparticle colloids in Brownian motion)[17]. (iii) Practical difficulties in fabricating precise nanostructures and locating biomolecules in the highly confined local electromagnetic field at the nanoscale[3].

Recently, there has been an emerging trend to integrate optical trapping and optical plasmonic trapping techniques in SERS detection systems[13,18-27], which enables the mechanical manipulation of nanostructured substrates in solution environments to improve the measurement reproducibility, stability, and efficiency[17,28]. Optical tweezers utilize the tightly focused laser beam to exert optical force to attract objects at focus against Brownian motion[29]. The position control of the trapped objects can be further enhanced and confined near plasmonic metallic nanostructures beyond the diffraction limit of the laser beam, known as optical plasmonic trapping[28-32], which is a promising tool to address the above challenges. Whereas the accessibility of existing optical plasmonic trapping-integrated SERS platforms is limited by the complicated fabrication of precise nanostructures, the special instrument to locate plasmonic nanocavity, and the potential perturbation of molecular immobilizations on the trapped objects. Hence, it awaits further developments to exploit the power of optical plasmonic trapping to facilitate the single-molecule SERS detections in aqueous milieus.

Here, we developed a convenient optical plasmonic tweezer-coupled SERS platform. After verifying single-molecule sensitivity, we studied the pH-dependent structural transition of tyrosine and the conformational features of hIAPP species under physiological pH conditions. With high-throughput capability, this platform holds the promise to advance the solution-phase single-molecule vibrational characterization method developments to study a biomolecule among heterogeneous mixtures during complex physiological processes.

## Results

### Creating dynamic nanocavity by optical plasmonic trapping for efficient single-molecule SERS detections in solutions

To develop an optical plasmonic tweezer-coupled SERS platform, we constructed a simple plasmonic junction between two AgNP-coated silica microbeads and placed it at the focus of two laser beams to trap an AgNP, which could form a SERS-active nanocavity among the trapped AgNP and the coated AgNP as illustrated in Fig. 1a. Supplementary Fig. 1 shows the experimental set-up combing a 532 nm

(2.7 mW) excitation laser and a 1064 nm trapping laser (8.0 mW) via stereo double-layer-pathways in a microscope for simultaneous optical manipulations and spectroscopic measurements. First, silica microbeads were functionalized with amino groups to chemically coat AgNP in sparse distribution (Supplementary Fig. 2), which were further bonded with the 5'-thiol-modified oligonucleotide L1 and the 5'-thiol-modified oligonucleotide L2, respectively. Next, the oligonucleotide L3 was added to tether L1 and L2 by complementary base pairing to generate AgNP-coated bead assemblies, giving a nanogap of around 20 nm. These bead assemblies were then deposited on the bottom of a microfluidic chamber, as visualized in the microscopic images of Fig. 1b (zoom-in) and Supplementary Fig. 3 (zoom-out). Prior to spectroscopic measurements, the analyte solution, the buffer solution, and the AgNP solution were injected into the three adjacent channels in the microfluidic chamber under laminar flow, which was maintained by adjusting the fluid flow rates in each channel to be equivalent. Upon cessation of the flow, analyte molecules and AgNPs were allowed to diffuse freely throughout the microfluidic chamber. Upon laser irradiation, a freely diffusing AgNP could be trapped at the plasmonic junction of an AgNP-coated bead dimer to form a dynamic nanocavity among the trapped AgNP and the coated AgNP, sandwiching the analyte molecules in the confined detection volume with a further enhanced local field for sensitive SERS characterizations. Moreover, the AgNP could be trapped with the 1064 nm trapping laser (on state) and released without it (off state), which enables dynamic switching of the trapped AgNP for high-throughput sampling. (Fig. 1).

As a proof of concept, we switched the 1064 nm trapping laser between on and off states and utilized the 532 nm excitation laser to conduct continuous SERS measurements at the plasmonic junction of an AgNP-coated bead dimer in 500 nM Nile Blue A (NBA) solution, where the formation of dynamic nanocavity could be controlled by the optical plasmonic trapping of an AgNP in close contact of the coated AgNP at the junction in Supplementary Fig. 4. By taking the time-averaged SERS spectra at each switching state, Fig. 1c demonstrates a regular pattern that the characteristic peak of NBA at ~1645 cm$^{-1}$ (ring stretching)[33] was emerged at on states and vanished at off states, giving the relative standard deviations (RSD) within 20% (Supplementary Fig. 5). Source data are provided in the Source Data file. Numerical simulation using FDTD method was carried out (Supplementary Note.1), which shows the electric field distribution of the plasmonic junction from the AgNP-coated bead dimer upon excitation in Supplementary Fig. 6. Since the trapping force is proportional to the square of electric field in Rayleigh regime[21], the trapped AgNP is confined against the coated AgNP in the plasmonic junction with the action force of optical plasmonic trapping and the reaction force at the contact surface as shown in Fig. 1d, giving a time-averaged total force of 8.8 pN upon excitation. Moreover, the trapping potential distribution in z-axis in Fig. 1d indicates the depth of the potential well at the equilibrium position as $3.5 \times 10^{-18}$ J, which is sufficient to overcome Brownian motion ($k_B T = 4 \times 10^{-21}$ J at room temperature) for stable AgNP trapping[29] and avoiding potential thermal damages[34]. More importantly, this platform enables the mechanical confinement of the trapped AgNP at a nanoscale precision since it is no longer subject to the optical diffraction limit. As shown in Fig. 1e, the dynamic nanocavity significantly enhances the localized electric field and provides the enhancement factors up to 10$^9$, which is adequate to detect a single molecule[35]. In comparison with existing optical plasmonic trapping experiments[13,22-24,26-28,32,36], this platform is easy to fabricate and locate without relying on expensive instruments, empowering high-throughput single-molecule SERS detections of freely diffusing samples in aqueous conditions.

To verify the single-molecule sensitivity and identify a single molecule in aqueous environments, we exploited bi-analytes SERS approach (BiASERS)[37] in the solution phase on the optical plasmonic

tweezer-coupled SERS platform. Methylene Blue (MB) and NBA were chosen as the bi-analytes partner because of their comparable Raman cross-section and fair detection probability under 532 nm excitation. We switched the 1064 nm trapping laser between on and off states and utilized the 532 nm excitation laser to conduct the SERS measurements at the plasmonic junction of an AgNP-coated bead dimer. Figure 2a displays the spectral mapping out of 3600 SERS measurements from twenty parallel experimental sessions collected at the dynamic nanocavity in time series, showing the distinct spectral features in the regions of 590-610 cm$^{-1}$ and 1620-1660 cm$^{-1}$. Statistical analysis of the data indicates that ~5 % spectra exhibited peaks intensities above three times the standard deviation of noise (Supplementary Fig. 7) and classified into the catalog of single-MB event, single-NBA event, or dual-MB and NBA event, based on the characteristic Raman peaks of MB or NBA at a high concentration of 10$^{-5}$M in Supplementary Fig. 8. Specifically, the appearance of the Raman peak at 1650 cm$^{-1}$ accompanied by the emergence of a peak at 600 cm$^{-1}$ represents the detection of single-NBA event (Fig. 2b blue), while the appearance of a peak at 1630 cm$^{-1}$ and the vanish of the peak at 600 cm$^{-1}$ represents the detection of single-MB event (Fig. 2b red)[33]. dual-MB and NBA event refers to the presence of the dual peaks at around 1640 cm$^{-1}$ (Fig. 2b black). Representative full spectra of each event are shown in Supplementary Fig. 9, and the fluctuation of peaks position was analyzed in Supplementary Fig. 10 to facilitate accurate identification among different events. The statistical histogram in Fig. 2c is predominated by single-dye-event (NBA or MB), suggesting that the signals were highly likely originated from a single molecule at the dynamic nanocavity. Furthermore, we repeated the BiASERS experiments by increasing the concentration of MB by

10 times (10$^{-7}$M) while keeping the concentration of NBA (10$^{-8}$M) unchanged. Most of the spectra contain the characteristic peaks of MB at 1630 cm$^{-1}$ with a large number of MB as the overflowing background molecules. Nevertheless, we still captured a few spectra in which the subtle peaks of NBA at 1650 cm$^{-1}$ emerged from the predominant peaks of MB at 1630 cm$^{-1}$ (Fig. 2 d), implying the presence of an NBA molecule in the MB-dominated nanocavity[38]. As the low-populated species, NBA was differentiated from the dominating MB at the single-molecule level.

## Monitoring the dynamic pH-dependent structures of tyrosine in solution

To further verify the aqueous stability and compatibility, we employed the optical plasmonic tweezer-coupled SERS platform to characterize tyrosine (Tyr) in different pH environments, considering that the side chains of numerous constituent amino acids interact to facilitate protein folding while their hydrogen bonding and electrostatic interactions are affected by environmental factors, such as pH[39]. The real-time SERS spectra of Tyr were monitored during the change from acidic to basic conditions in Fig. 3a. First, the microfluidic chamber was filled with 50 μM Tyr solution at pH 1.0, generating the characteristic peak at 1620 cm$^{-1}$ as the in-plane ring stretching ($\nu_{8a}$ mode) of Tyr at +1 charged state[40]. Then 2 M NaOH was added through a side channel to adjust the environmental pH from 1.0 to 13.0, which was measured at the outlet of the microfluidic chamber by pH indicator strips and verified by stoichiometric calculations. Meanwhile, the corresponding SERS spectra were collected continuously. Source data are provided in the Source Data file. There is a downshift of $\nu_{8a}$ mode to 1602 cm$^{-1}$ along with the gradual conversion of Tyr to −2 charged state[40] and

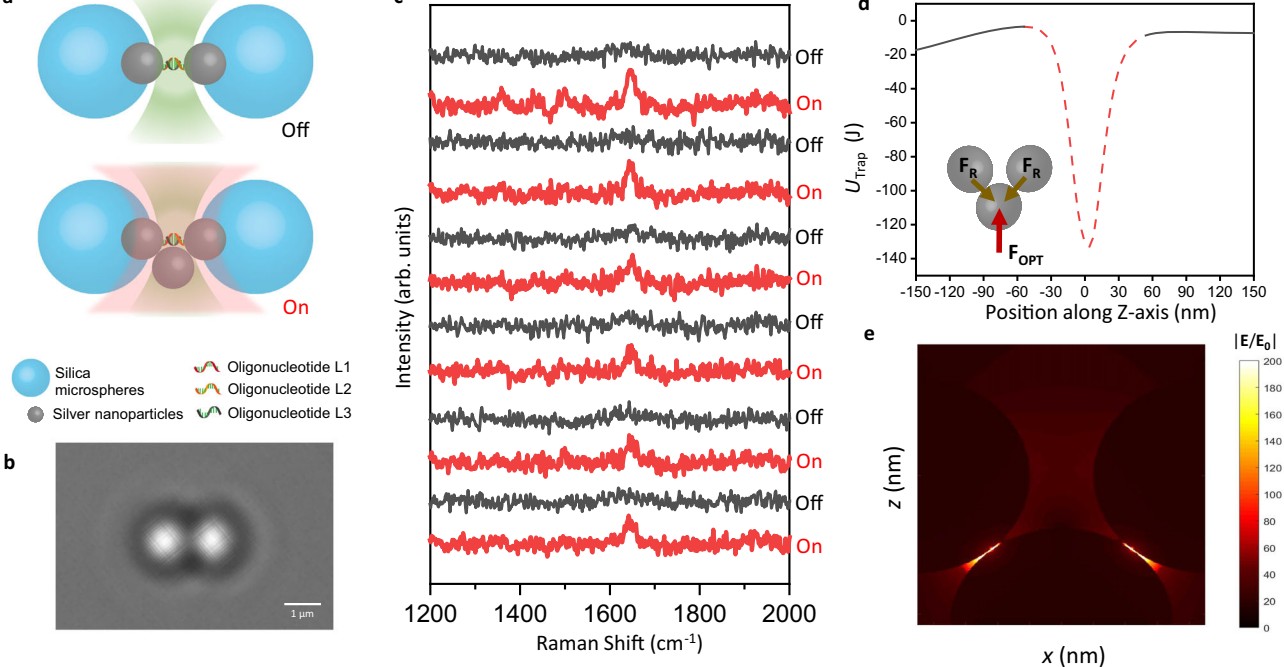

**Fig. 1 | Dynamic nanocavity controlled by optical plasmonic trapping.**
**a** Schematic illustration of the optical plasmonic tweezer-coupled SERS platform. Top: Off-state that has the 532 nm excitation laser (green) for SERS measurements. Bottom: On state that has the 532 nm excitation laser (green) and the 1064 nm trapping laser (red) to form the dynamic nanocavity among a trapped AgNP and two coated AgNP at the plasmonic junction of an AgNP-coated microbead dimer. **b** Brightfield image of an oligonucleotide-linked AgNP-coated silica microbead dimer by a conventional optical microscope. The scale bar is 1 μm. The micrograph is representative of at least three independent experiments. **c** The time-averaged SERS spectra over 5 s recorded at an AgNP-coated bead dimer in 500 nM NBA

solution by the 532 nm excitation laser with the switching of the 1064 nm trapping laser between on (red) and off (black) states for 12 times. The term (arb. units) is abbreviated for arbitrary units. **d** Calculated trapping potential along the z-axis. The red dashed line represents the virtual trapping potential within the plasmonic junction of an AgNP-coated microbead dimer, where the trapped AgNP is confined against the coated AgNP. Insert: The confinement of the trapped AgNP due to a balance between the action force of optical plasmonic trapping (**F$_{OPT}$**) and the reaction force at the contact surface (**F$_R$**). **e** FDTD simulation of the **E**-field (|**E**/**E$_0$**|) distribution at the dynamic nanocavity among the trapped AgNP and the coated AgNP in the plasmonic junction.

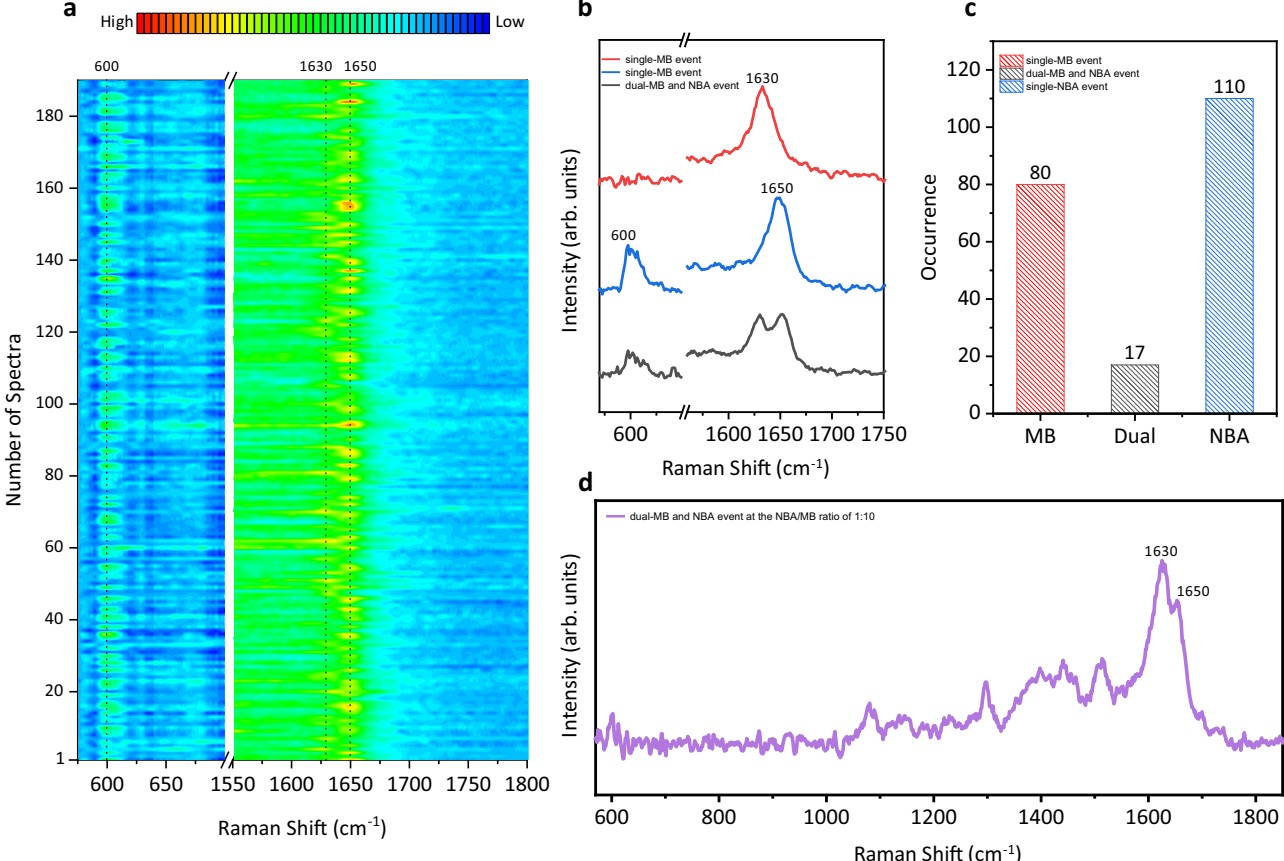

**Fig. 2 | BiASERS analysis of MB and NBA at dynamic nanocavity. a** Spectral mapping of $10^{-8}$ M MB and NBA mixture solution from 3600 SERS measurements across twenty parallel sessions. Integration time: 1 s per spectrum. Only the spectra with the peak intensities above three times the standard deviation of the noise are shown. **b** Representative spectra of single-MB event (red), single-NBA event (blue), and dual-MB and NBA event (black). **c** Histogram of single-MB event, single-NBA event, and dual-MB and NBA event from the acquired spectra. **d** Representative spectrum of dual-MB and NBA event showing the signal of low-populated NBA emerged from the predominant signal of MB at the NBA/MB ratio of 1:10. The term (arb. units) is abbreviated for arbitrary units.

a change in the intensity ratio of Tyr doublet located at ~854 cm$^{-1}$ and ~830 cm$^{-1}$, which is assigned to the Fermi resonance between the symmetric ring-breathing ($v_1$) and the overtone of the out-of-plane ring deformation ($2v_{16a}$)[41]. The intensity of 830 cm$^{-1}$ increases and the ratio of $I_{830}/I_{854}$ changes from approximately 1:1 to 2:1[41,42]. These differences are attributed to the increase of negative charges on the phenolic hydroxyl group (pKa=10.5) upon the adjustment from acidic to basic conditions[41]. Other changes, such as the increasing intensity of the peak at 1071 cm$^{-1}$ (C-N stretching) and the peak shifting from 1153 cm$^{-1}$ to 1161 cm$^{-1}$ (C-C-N asymmetric stretching), are also observed and consistent with literatures[43]. On the other hand, there is no obvious signal at 930 cm$^{-1}$ (C-COO$^-$ stretching), 1390 cm$^{-1}$ (COO$^-$ symmetric stretching), and 1090 cm$^{-1}$ (amino group vibration) related to Ag-Tyr interactions via the carboxylate and amino groups[44], owing to the attachment of the nucleotides on the AgNP surface as a protected layer for Tyr. Moreover, at the midpoint of the pH adjustment process, the spectra measured with a long integration time (10 s) show the spectral broadening of $v_{8a}$ mode at 1610 cm$^{-1}$ in Fig. 3a (green), while the spectra acquired with a short integration time (1 s) exhibit sharp and fluctuated peaks in the range of 1620 to 1602 cm$^{-1}$ in Fig. 3b, indicating the dynamic snapshots of single or a few Tyr molecules with distinct charge states at the dynamic nanocavity analogues to the previous single-molecule SERS studies[44].

To understand the chemical environment of Tyr in aqueous solutions, we compared the SERS spectra of Tyr with the calculated Raman spectra of Tyr at different charge states in Fig. 3c. Using density functional theory (DFT) simulations, the construction of Tyr-H2O

clusters was based on Ghomi and co-workers' model by involving seven water molecules in the vicinity to coordinate with the amino group (pKa=9.2), the carboxyl group (pKa=2.2), and the phenol hydroxyl group (pKa = 10.5) of Tyr (Supplementary Note. 2, Supplementary Fig. 11)[45], then their structures were further optimized at +1 and −2 charged states to match the experimental results at pH 1.0 and pH 13.0, respectively. Due to the inherent deficiencies related to DFT simulations (i.e. neglect of anharmonicity, incomplete incorporation of electron correlation, and other approximations), the calculated frequencies were overestimated especially in higher wavenumber region[46,47]. Based on the least-squares procedures[48,49] (Supplementary Note. 3, Supplementary Figs. 12 and 13), an empirical scaling factor 0.9770 was applied to the wavenumbers above 1000 cm$^{-1}$ in the simulated Raman spectra of Tyr at different charge states to align with the experimental observations at pH 1.0 and pH 13.0, respectively. Supplementary Fig. 11 presents possible hydrogen-bonds between Tyr and water molecules and the charges of Tyr nitrogen and oxygen accordingly. The electrostatic potential (ESP) mapping[50] in Fig. 3d, e demonstrate the charge distribution of Tyr with the electrophilic moieties in red and the nucleophilic moieties in blue. These results indicate that both the phenol hydroxyl group and the carboxyl group of Tyr become negatively charged after the deprotonation at pH 13.0, which depolarizes the whole molecule and increases the Raman activity of in-plane ring stretching to account for the downshift and enhancement of $v_{8a}$ mode[42,43]. Moreover, the hydroxyl group serves as both a hydrogen-bond donor and acceptor in acidic and neutral environments, while its oxygen is exposed as a sole hydrogen-bond

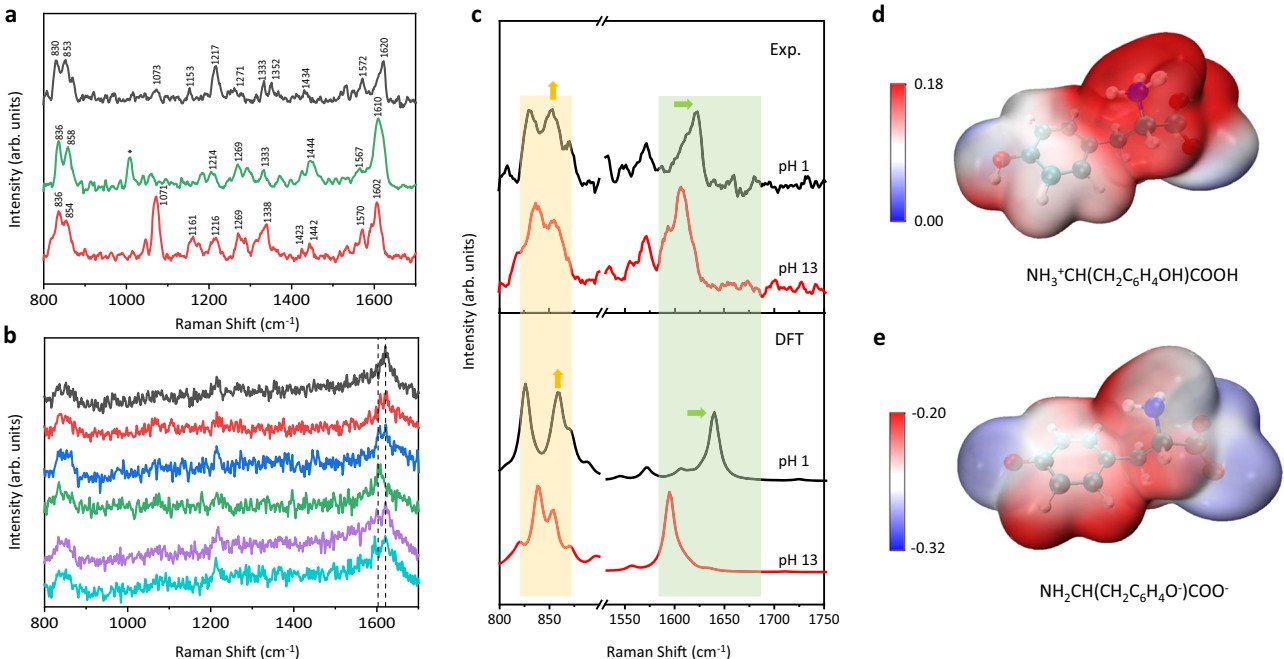

**Fig. 3 | pH-dependent structural transitions of Tyr. a** Real-time SERS spectra of Tyr in the condition changes from pH 1.0 (black) to pH 10.5 (green), and pH 13.0 (red). Integration time: 10 s. **b** Representative SERS spectra of Tyr at pH 10.5 with 1 s integration time, showing spectral fluctuations arisen from Tyr in distinct charge states. **c** Comparison of SERS spectra (upper panel) and simulated Raman spectra (lower panel) of Tyr at pH 1.0 (black) and pH 13.0 (red), respectively. The SERS spectra are adopted from (**a**). The simulated Raman spectra and electrostatic potential are generated from DFT calculations on Tyr in +1 charged state as $NH_3^+CH(CH_2C_6H_4OH)COOH$ (**d**) and -2 charged state as $NH_2CH(CH_2C_6H_4O^-)COO^-$ (**e**), respectively. Based on least-squares procedures, a scaling factor of 0.9770 was applied to the wavenumbers above 1000 cm$^{-1}$ in simulated Raman spectra. The term (arb. units) is abbreviated for arbitrary units. **d, e** Electrostatic potential (ESP) values of Tyr in +1 charged state and -2 charged state mapped on 0.01 a.u. van der Waals surface, expressed in atomic units (a.u.).

acceptor under basic conditions. The loss of hydrogen and the increase in negative charges on phenolic oxygen lead to the enhanced intensity at 836 cm$^{-1}$ in the Tyr doublet[41].

## Characterizing transient species of hIAPP under different physiological pH

With efficiently reduced detection volume and integration time to overcome the ensemble and temporal averaging of heterogeneous species[25], we utilized the optical plasmonic tweezer-coupled SERS platform to probe the structures of various hIAPP species at two physiological pH conditions (pH 5.5 and pH 7.4)[25]. As an amyloidogenic IDP, free hIAPP monomers possess the propensity to assemble into heterogeneous aggregates and amyloid fibrils[8]. Its ensemble oligomers in dilute solutions and amyloid fibrils in solid forms are well characterized by SERS studies[51,52]. Yet, it is unclear how the environmental factors affect the formation of different hIAPP species to determine aggregation pathways[5]. Specifically, the unclear molecular mechanisms underlying the documented inhibitory effect of acidic pH on fibrillation, in contrast to the prevalent utilization of neutral pH to promote amyloid aggregation[5], await further investigations.

At pH 5.5 condition, hIAPP has four positive charged sites (α-amino group, Lys-1, Arg-11, and His-18), as illustrated in Fig. 4a. The CD spectra of 10 μM hIAPP in the PBS buffer of pH 5.5 for incubation at $t = 0$, 2, and 24 h present a persistent negative peak at around 203 nm with two shoulders at 212 and 223 nm in Fig. 4b (black, green, blue), which are consistent with previous studies[53] to confirm the ensemble conformation as major random coils and minor α-helix[5]. This concentration mimics a near-physiological milieu involving moderate intra- and inter-molecular interactions that are crucial to initiate the aggregation process and stabilize the transient intermediates of hIAPP[53]. Although it is below the detection threshold of

spontaneous Raman spectroscopy (Supplementary Fig. 14), we have obtained the clear SERS signals of hIAPP on the optical plasmonic tweezer-coupled SERS platform. To reflect ensemble structures, 10000 SERS spectra were collected at each early timepoint during the incubations of hIAPP from different batches of independent preparations in multiple parallel experimental sessions. Source data are provided in the Source Data file. Figure 4c−f show the representative SERS spectra of 10 μM hIAPP solution under pH 5.5 incubation at $t = 0$ h (black) and $t = 2$ h (green). The characteristic peaks are assigned to protein backbones including amide I band (1656 cm$^{-1}$), CH$_2$ deformation (1450 cm$^{-1}$) and amide III band (1250 cm$^{-1}$ and 1287 cm$^{-1}$), as well as specific residues such as Phe (1006 cm$^{-1}$, 1585 cm$^{-1}$), Tyr (830 cm$^{-1}$ and 850 cm$^{-1}$, 1605 cm$^{-1}$), and Cys-Cys (523 cm$^{-1}$)[54]. The details of peak assignments are listed in Supplementary Table. 1. To better analyze protein secondary structures, we conducted the second derivative analysis[54] for the amide I region from 1550 cm$^{-1}$ to 1750 cm$^{-1}$ in Fig. 4f to plot Fig. 4g, and presented the mapping of these secondary derivative spectra in Supplementary Fig. 15 ($t = 0$ h) and Fig. 4h ($t = 2$ h). The spectra at $t = 0$ and $t = 2$ h exhibit the identical amide I band centered at 1656 cm$^{-1}$, which are attributed to the helical and unstructured hIAPP[51,52]. It is further supported by the amide III bands at 1250 cm$^{-1}$ and 1287 cm$^{-1}$, assigned to random coil and α-helix structures[39,51] respectively. In addition, the peak at 523 cm$^{-1}$ represents the gauche-gauche-trans/trans-gauche-gauche (g-g-t/t-g-g) conformation of the disulfide bridge between Cys2 and Cys7[54]. The Tyr doublets with the value of $I_{830}/I_{854} ≈ 1$ in the SERS spectra of hIAPP under pH 5.5 (Fig. 4e) imply that the C-terminal Tyr residue acted as both hydrogen-bond donor and acceptor in exposure to water molecules[41,54], in consistent with the extended and coil-rich conformations of hIAPP reported in the physiological acidic environment[5]. Furthermore, MD simulations were conducted to monitor the monomeric conformational dynamics of

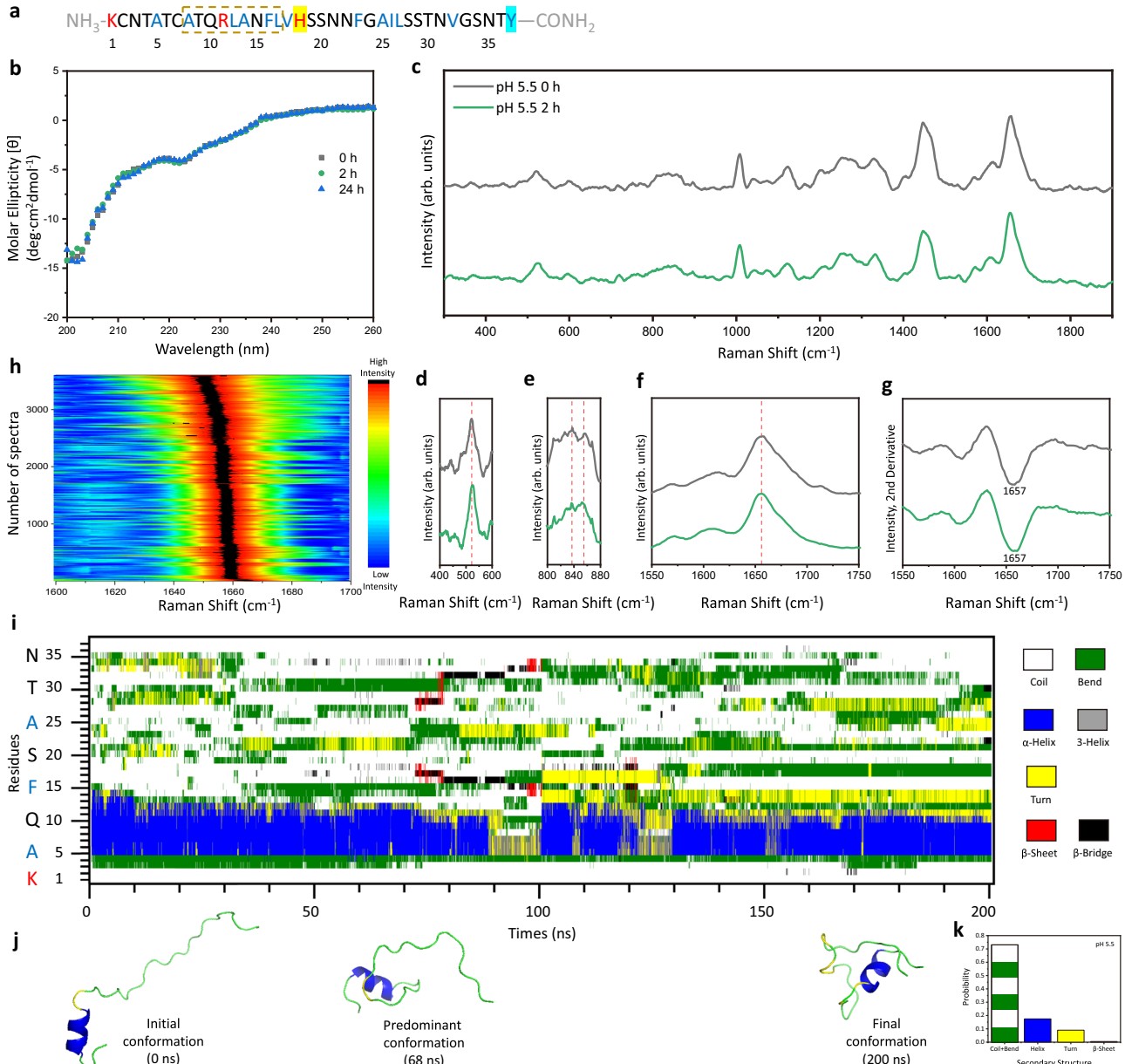

**Fig. 4 | Structural characterizations of hIAPP incubated under pH 5.5.**
**a** Properties of hIAPP residues with +4 charges at pH 5.5: Positively charged residues in red, hydrophobic residues in blue, and residues with helical potency in the dashed box. **b** CD spectra of 10 μM hIAPP solution under pH 5.5 incubation at $t = 0$, 2, and 24 h. **c** Representative SERS spectra of 10 μM hIAPP solution under pH 5.5 incubation at $t = 0$ h (black) and $t = 2$ h (green). **d** The disulfide bond region of (**c**). **e** The Tyr doublets region of (**c**). **f** The amide I region of (**c**). **g** Secondary derivative spectra of (**f**). The term (arb. units) is abbreviated for arbitrary units. Representative spectra were observed more than 10 times in parallel SERS measurements.

**h** Mapping of secondary derivative spectra of the amide I region of the SERS spectra of hIAPP at $t = 2$ h incubation under pH 5.5 from parallel experimental sessions. The color bar shows the normalized intensities from low (dark blue) to high (red-black). **i** Representative time evolution of secondary structure per residue of monomeric hIAPP with +4 charges at pH 5.5. Color legends for secondary structures: White: Coil; Green: Bend; Blue: α-Helix; Grey: 3-Helix; Yellow: Turn; Red: β-Sheet; Black: β-Bridge. **j** Snapshots of the initial conformation (0 ns), predominant conformation (68 ns), and final conformation (200 ns) from (**i**). **k** Statistical analysis of the secondary structural contents of the monomeric hIAPP from (**i**).

hIAPP at +4 charged state under pH 5.5 condition, where the initial input was modified from Hansmann and co-workers' unstructured hIAPP model[55] to reflect experimental observations. In Fig. 4i, k, the secondary structure evolution and the statistics of structural contents show the extensive disordered structures with little helix and turn segments of hIAPP at pH 5.5, consistent with the CD and SERS spectra for its bulk assessments. Specifically, the predominating hIAPP conformation adopts mainly random coils with a short helix near N-terminal, as illustrated in Fig. 4j. The high conformational flexibility and electrostatic repulsions could give rise to the swollen coil conformation and plausible inhibitory effect on fibril development[5,56], as

proven by the persisting predominant helix-coil structures in the CD and SERS of hIAPP after the 24-hour incubation at pH 5.5.

At pH 7.4 condition, His-18 (pK$_{a3}$ = 6.0) of hIAPP is deprotonated to reduces the positive charged sites to three, as illustrated in Fig. 5a. In Fig. 5b, the CD spectra of 10 μM hIAPP under pH 7.4 incubation at $t = 0$ h and $t = 2$ h (black and green) show a peak at 203 nm with two shoulders at 212 and 223 nm to imply the ensemble conformation with major random coil and minor helix structures at the early stage of aggregation, followed by a dramatic change into a strong peak at 218 nm at $t = 24$ h (blue) to suggest the formation of amyloid fibrils with β-sheet structure at the later stage of aggregation[5]. Focusing on

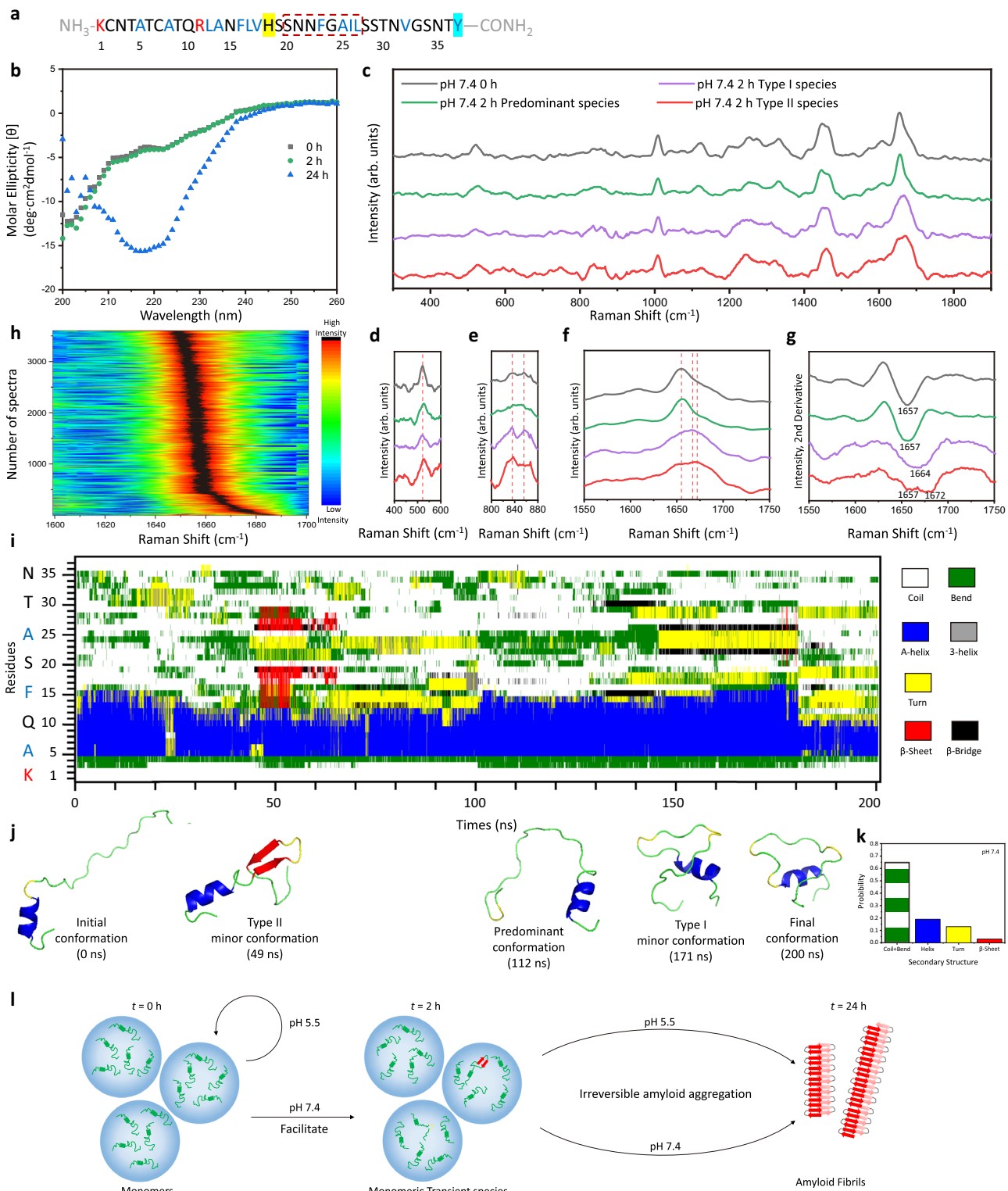

the early stage of aggregation, the representative SERS spectra of 10 μM hIAPP under pH 7.4 incubation at $t = 0$ h and $t = 2$ h are shown in Fig. 5c, along with the plots of the disulfide bond region (Fig. 5d), the Tyr doublets region (Fig. 5e), the amide I region (Fig. 5f) and the corresponding second derivative spectra (Fig. 5g), respectively. Source data are provided in the Source Data file. At $t = 0$ h, the spectrum of hIAPP at pH 7.4 (black) demonstrates the amide I band at 1656 cm$^{-1}$, which is attributed to the helix-coil structure and similar to the initial conformation at pH 5.5. (More spectra are shown in Supplementary Fig. 16.) However, after 2-hour incubation at pH 7.4, we

observed emerging sub-populations of SERS spectra of hIAPP from a statistically significant sampling ($n = 10000$) based on the efficient measurements at the dynamic nanocavity with 1 s accumulation time in multiple parallel experimental sessions, as indicated by the spectral mapping in Fig. 5h. The most frequently appeared spectrum (Fig. 5, green) exhibits a typical amide I band at 1656 cm$^{-1}$, representing the predominant helix-coil structure of hIAPP at 7.4. Intriguingly, two distinct types of spectra were obtained occasionally and reproducibly, as shown in Fig. 5, purple and red. The type I of rare spectra (purple) features the broad amide I band at 1668 cm$^{-1}$,

**Fig. 5 | Structural characterizations of hIAPP incubated under pH 7.4.**
**a** Properties of hIAPP residues with +3 charges at pH 7.4: Positively charged residues in red, hydrophobic residues in blue, and His−18 highlighted in yellow as the deprotonated residue upon the change of pH from 5.5 to 7.4. **b** CD spectra of 10 μM hIAPP solution under pH 7.4 incubation at $t$ = 0, 2, and 24 h. **c** Representative SERS spectra of 10 μM hIAPP solution under pH 7.4 incubation at $t$ = 0 h (black) and $t$ = 2 h (most-populated spectrum, green; type I rare spectrum, purple; type II rare spectrum, red). **d** The disulfide bond region of (**c**). **e** The Tyr doublets region of (**c**). **f** The amide I region of (**c**). **g** Secondary derivative spectra of (**f**). The term (arb. units) is abbreviated for arbitrary units. Representative spectra were observed more than 10 times in parallel SERS measurements. **h** Mapping of secondary derivative spectra of the amide I region of the SERS spectra of hIAPP at $t$ = 2 h incubation under pH 7.4 from parallel experimental sessions. The color bar shows the normalized intensities from low (dark blue) to high (red-black). **i** Representative time evolution of secondary structure per residue of monomeric hIAPP with +3 charges at pH 7.4. Color legends for secondary structures: White: Coil; Green: Bend; Blue: α-Helix; Grey: 3-Helix; Yellow: Turn; Red: β-Sheet; Black: β-Bridge. **j** Snapshots of the initial conformation (0 ns), type II minor conformation (49 ns), predominant conformation (112 ns), type I minor conformation (171 ns), and final conformation (200 ns) from (**i**). **k** Statistical analysis of the secondary structural contents of the monomeric hIAPP from (**i**). **l** Cartoon illustration of the hIAPP amyloid developments affected by physiological pH environments (pH 5.5 and pH 7.4).

attributed to an arising turn structure[57]. Moreover, the intensity ratio of the Tyr doublets $I_{830}/I_{854} \approx 1$ in type I of rare spectra indicates the exposure of the C-terminal Tyr-37 in an extended conformation[41,54]. The type II of rare spectra (red) shows the characteristic amide I band at 1655 and 1674 cm$^{-1}$, implying the co-existence of helix-coil and β-sheet structures[51,52]. In addition, the Tyr doublets with the intensity ratio $I_{830}/I_{854} > 1$ suggest that the C-terminal Tyr-37 slightly shifted toward hydrogen donating as interrelated to its surrounding chemical environment[54,58]. As evident from these spectral features, the type I spectra might be putatively assigned to the transient species of hIAPP containing the turn structure with an extended C-terminal, while the type II spectra might be assigned to the transient species containing both helix-coil and β-sheet structures with a constrained C-terminal. Notably, the spectra of type I and type II species both contain a peak at 523 cm$^{-1}$, indicative of the intact disulfide bond to stabilize the N-terminal helix structure. There is no obvious peak at 490 cm$^{-1}$, which is related to disulfide bond restrictions (Supplementary Fig. 17), implying these hIAPP species in primary monomeric or low-order oligomeric forms without forming compact aggregates or amyloid fibrils[54]. MD simulations of the monomeric hIAPP at pH 7.4 demonstrate the emerging turn and β-sheet fractions among the predominating helix and coil structures in the secondary structure evolution and content in Fig. 5i and k. Top three most-populated conformations are illustrated in Fig. 5j. The predominant conformation exhibits the partial helix-coil structure. The type I minor conformation has a turn at residues 20–25. The type II minor conformation contains a short β-hairpin structure (residues 14–19 and 26–29), giving lower values of the root-mean-square fluctuation (RMSF) at residues 30-37 than that of the type I conformer to reflect a more constrained C-terminal (Supplementary Fig. 18). These results are in good agreement with the experimental observations and consistent with previous MD simulations[59–61], providing molecular details of the monomeric species that might contribute to the type I and II spectra of hIAPP at the very early aggregation stage. In addition, since His-18 (pK$_{a3}$ = 6.0) deprotonated at pH 7.4, it reduces electrostatic repulsion to promote intra- and inter-molecular interactions[56]. As indicated by the distinct change of Tyr doublets in the spectra of type II species (red), Tyr-37 might interact with His−18 by hydrogen bonding and/or ring stacking to constrain C-terminal, which could further stabilize the β-sheet structure in type II transient species and drive the subsequent aggregation[42,62]. This assignment is supported by the previous MD simulations that the interaction between His−18 and Tyr-37 would minimize the entropic cost of initial oligomerization[63] and facilitate the intermolecular contact[60] upon aromatic stacking and hydrophobic collapse[59,64] to assemble β-sheet-rich structures in low-order oligomers[59,60,63,64]. Although existing in a low population, the transient species of hIAPP were directly differentiated from the dominating helix-coil species at the early lag phase of the amyloid formation at pH 7.4, which reveals the influence of pH on the structural conversions of hIAPP and demonstrates the effectiveness of our SERS platform to characterize a few heterogeneous proteins involving in molecular interactions under physiological pH environments.

For comparison, we incubated 10 μM hIAPP at pH 7.4 for an initial period of 2 hours, followed by a subsequent adjustment of pH from 7.4 to 5.5 to continue the incubation for the succeeding 22 hours. Interestingly, as shown in the CD spectra in Supplementary Fig. 19, a strong peak at 218 nm was emerged at $t$ = 24 h (total incubation time), attributed to the β-sheet structure. AFM images confirm the presence of amyloid fibrils (Supplementary Fig. 20). No obvious SERS signal of hIAPP was detected at the nanocavity, due to the size limit[51]. We then utilized AgNP colloids to perform the bulk SERS measurement of these amyloid fibrils generated after the 24-hour incubation involved the pH adjustment from 7.4 to 5.5 at $t$ = 2 h. Their SERS spectrum in Supplementary Fig. 21 exhibits the amide I band at 1674 cm$^{-1}$ and amide III band at 1226 cm$^{-1}$ indicative of the ordered β-sheet structure[52] and the peak at 490 cm$^{-1}$ attributed to the strained disulfide bond[54], which are identical to the characteristic features of typical hIAPP fibrils with compact β-sheet structures generated from the 24-hour incubation at pH 7.4 (Supplementary Fig. 22). These results confirm the formation of hIAPP amyloid fibrils despite the pH adjustment to 5.5 for the succeeding incubation. It implies that the critical species of hIAPP might have arisen during the initial 2-hour incubation at pH 7.4, thereby facilitating subsequent amyloid aggregations irrespective of the pH conditions for continued incubations. In particular, we identified two types of low-populated transient species at the early stage of hIAPP aggregation under pH 7.4, which are consistent with the β-sheet-containing oligomers observed in previous SERS, 2D-IR, and MD simulation studies[59,60,64]. Specifically, the type I species possesses a turn structure (residues 20-25), which is counted as a critical loop-forming region to promote aggregation[64]. The type II species contains a short β-hairpin (residues 14-19 and 26-29), in accordance with the proposed amyloidogenic precursors and building blocks of amyloid fibrils in the literatures[59,60]. Furthermore, it is reported that significant changes of conformational populations occurred in the early stage as the development from monomers to oligomers[59,60], followed by the increase of oligomeric structures persisting the complementary of turns and β-sheets in a partially ordered−disordered arrangement[60] to form amyloid fibrils. Hence, the direct characterizations of these early-stage transient species are crucial to resolve the amyloid aggregation of hIAPP. Figure 5l illustrates the influence of pH on the amyloid development of hIAPP. The acidic environment significantly retards the structural conversion of hIAPP and preserves the helix-coil conformations for 24 h. Whereas, the neutral environment maintains the predominant helix-coil structures but induces the formation of the low-populated transient species of hIAPP, resulting in the growth of amyloid fibrils. These pH-regulated transient species might slightly shift the dynamic conversions of hIAPP species into amyloid-competent and alternative conformations among the predominant helix-coil structures to promote amyloid fibril formations even after adjusting back to acidic conditions. Under different pH conditions, the protonated/deprotonated state of His-18 influences its electrostatic interactions for structural conversions. While the C-terminal residues also facilitate the partially folded transient species towards irreversible amyloid aggregation[63]. Thus, the previous strategies that focused on a single site of hIAPP

might not be optimal in blocking amyloid aggregations, while targeting its on-pathway transient species could be a promising strategy[4,65,66].

## Discussion

In summary, we utilized optical plasmonic trapping to construct a dynamic nanocavity, which reduces the diffraction-limited detection volume and generates reproducible SERS enhancements for efficient single-molecule characterizations in solution. By switching the trapping laser between on and off states, an AgNP was trapped and released at the plasmonic junction of an AgNP-coated silica microbead dimer, respectively, which enables efficiently and continuously high-throughput detections at the well-defined location under microscopic visualization. As verified by BiASERS experiments, this optical plasmonic tweezer-coupled SERS platform could identify the low-populated species among overflowing background molecules with single-molecule sensitivity. Moreover, we monitored the real-time structural transitions of Tyr upon the changing pH conditions and analyzed its pH-dependent vibrational characters from two distinct charge states as an environmental indicator. Furthermore, we characterized the conformations of hIAPP incubated under two physiological pH 5.5 and 7.4, where the different species of hIAPP were in a dynamic mixture and difficult to detect in previous ensemble and temporal averaging measurements[4,51]. Based on the statistically significant sampling on our efficient single-molecule SERS platform combined with MD simulations, two types of low-populated hIAPP transient species were differentiated from its predominant monomers at the early stage of pH-induced amyloid aggregation, containing the critical turn structure or the partial β-sheet with constrained C-terminal. Such a slight shift in the equilibrium between different hIAPP species stimulated irreversible amyloid development even after the post-adjustment of pH. Hence, the direct structural characterizations of the early-stage transient species of hIAPP among heterogeneous mixtures involving intra- and inter-molecular interactions would provide profound mechanistic insights to understanding the complex amyloid aggregation processes associated with type II diabetes.

This optical plasmonic tweezer-coupled SERS platform offers a strategy to address the challenge of characterizing a single molecule from heterogeneous mixtures in aqueous milieus[4,9]. Since both optical plasmonic trapping and SERS techniques are surface-sensitive relying on nanostructured substrates, the integration overcomes the optical diffraction limit to confine the position of the plasmonic nanocavity and reduce the SERS active-detection volume for consistently high SERS enhancements. Therefore, this platform can detect the freely diffusing analytes at the single-molecule level without molecular immobilization or solution dilution, holding great potential to unveil various molecular behaviors and interactions in complex biological processes.

## Methods

### Materials

Silica beads were purchased from Spherotech Inc. Silver nitrate (≥ 99.0%), trisodium citrate (≥ 99.0%), Tris (≥ 99.9%), sodium chloride (≥ 99.0%), (3-aminopropyl) triethoxysilane (≥ 98.0%), Nile blue A (≥ 99.0%), methylene blue (≥ 99.0%), and tyrosine (≥ 99.0%) were purchased from Sigma-Aldrich. hIAPP containing the disulfide bridge between Cys-2 and Cys-7 and the amidated C-terminal was purchased from GL Biochem (Shanghai) Ltd. Oligonucleotides of sequences 1, 2 and 3 (L1, L2, and L3) were customized from BGI Genomics (Hong Kong). L1 and L2 were functionalized with a thiol group (-SH) at 5' end.

The sequences of L1, L2, and L3 are as follows (from 5' to 3').

L1: HS-AAAAAAAAAAATCTCAACTCGTA
L2: HS-AAAAAAAAAACGCATTCAGGAT
L3: AGAGTTGAGCATATCCTGAATGCG

### Preparation of AgNP and AgNP-coated silica microbeads

AgNP were synthesized by adding 1.0 mL trisodium citrate solution (0.1 M) to 50 mL boiled AgNO$_3$ solution (1 mM) and boiling for 16 min under constant stirring. After cooling down to room temperature, the synthesized AgNP (70 nm) colloid was rinsed with Milli-Q water for three times to remove the excess reducing agent. Colloidal AgNP solutions were prepared with a concentration of $10^{-11}$ M at different pH, and their stabilities were assessed by dynamic light scattering (DLS) and zeta-potential measurements (Supplementary Fig. 23). To ensure the constant average hydrodynamic diameters and zeta-potential values reflecting the stability of AgNP, all colloidal AgNP solutions were freshly prepared and immediately used within 0.5 hour in parallel measurement sessions.

AgNP-coated silica beads were prepared according to the following protocols[25]. 1 mL silica beads (1.26 μm, 5.0% w/v) were dried overnight at 60 °C and redispersed in 500 μL anhydrous ethanol awaiting surface modification for AgNP coating. Next, 1 mL ethanol solution containing 0.2% (3-aminopropyl) triethoxysilane (APTES) was added to the bead suspension (final concentration 0.1%) and reacted for 24 h at room temperature under continuous stirring. The resulting solution was then purified by centrifuging with distilled ethanol at 1500 × g for three times and discarding the supernatant. The remaining pellet was further dried at 60 °C to remove the ethanol. Followed by adding 4 mL Milli-Q water to redisperse, the silica beads with the amino groups modified surface were obtained. Lastly, silica beads with sparsely and uniformly coated AgNP were prepared by adding the dispersed APTES-modified beads to the AgNP colloids in an AgNP colloids:APTES-modified silica beads ratio of 995:5 (v/v) under agitation and reacting for 10 min at room temperature.

### Construction of Oligonucleotide-linked AgNP-coated bead assembly

Thiolated oligonucleotides L1 and L2 were used to conjugate with AgNP-coated beads, while L3 with the complementary sequence to part of L1 and L2 was used for hybridization. A mixture of 13.5 μL SDS solution (1% v/v), 135 μL phosphate buffer (100 mM, pH 7.4), and 200 μL L1 or L2 solution (30 μM) was firstly added to 1 mL aforementioned AgNP-coated beads suspension. Over a period of 30 min, 110 μL NaCl solution (2 M) was added in a stepwise manner. After overnight incubation at room temperature, the solution was centrifuged three times (1500 × g), and the supernatant was removed to eliminate residual oligonucleotides and unmodified metal-coated silica beads. Then the precipitate was redispersed in 100 μL PBS buffer (10 mM phosphate, 150 mM NaCl, pH 7.4) to obtain the L1 or L2 conjugated AgNP-coated beads. Next, 100 μL of L1 conjugated and L2 conjugated AgNP-coated beads were mixed with the addition of 50 μL L3 solution (30 μM) to allow hybridization at room temperature overnight. Finally, the oligonucleotide-linked AgNP-coated bead assemblies were obtained. The construction of oligonucleotide-linked AgNP assemblies follows the same protocol except that the AgNP-coated beads suspensions were replaced with AgNP colloids.

### hIAPP preparation

hIAPP containing the disulfide bridge between Cys-2 and Cys-7 and the amidated C-terminal was purchased from GL Biochem (Shanghai) Ltd. It was dissolved in hexafluoro-isopropanol (HFIP) at a concentration of 1 mg/mL and incubated for 1 h, followed by lyophilization. Then, the purified hIAPP was rehydrated in PBS buffer (10 mM phosphate, 150 mM NaCl, pH 7.4) and filtered through a 0.22 μm Tuffryn syringe filter to prepare the stock solutions with a concentration of 1 mg/mL (250 μM). In parallel experimental sessions, several batches of the hIAPP stock solutions were prepared as independent samples for the subsequent incubations at different pH conditions. Portions of the hIAPP stock solutions were diluted with PBS buffer (10 mM phosphate, 150 mM NaCl, pH 5.5) to achieve a concentration of 10 μM and adjusted

to pH 5.5 with 1 M HCl (if necessary) under the monitoring of a pH meter, followed by the 24-hour incubation at 37 °C. Portions of the hIAPP stock solutions were diluted with PBS buffer (10 mM phosphate, 150 mM NaCl, pH 7.4) to achieve a concentration of 10 µM and adjusted to pH 7.4 with 1 M HCl/1 M NaOH (if necessary) under the monitoring of a pH meter, followed by the 24-hour incubation at 37 °C. Portions of the hIAPP stock solutions were diluted with PBS buffer (10 mM phosphate, 150 mM NaCl, pH 7.4) to achieve a concentration of 10 µM and adjusted to pH 7.4 with 1 M HCl/1 M NaOH (if necessary) under the monitoring of a pH meter, followed by the 2-hour incubation at 37 °C. Subsequently, the pH of this solution was further adjusted from pH 7.4 to 5.5 through the dropwise addition of 1 M HCl under the monitoring of a pH meter, followed by the succeeding 22–hour incubation at 37 °C. Meanwhile, the time-dependent CD, SERS, and AFM characterizations were conducted repeatedly for parallel experimental sessions during the incubations of these hIAPP solutions from different batches of independent preparations.

### Instrumental setup and dynamic SERS measurements

The instrumental setup is illustrated in Supplementary Fig. 1. A 532 nm excitation source (MLL-III–532-50 mW, CNI, China) was directed into a built-in optical tweezers system with a 1064 nm trapping laser and bright field microscope imaging system (LUMICKS, Netherlands). The 532 nm and 1064 nm laser beams were combined by a dichroic mirror and collimately focused into a microfluidic chamber through a 60× water immersion objective (Olympus, LumplanFLN60x, N.A. = 1.2). The backscattered light was reflected by a 750 nm long-pass dichroic mirror and filtered by a 532 nm notch filter before entering a spectrometer (IsoPlane SCT-320, 1200 lines/mm, Teledyn Princeton Instrument, United States) with a liquid nitrogen-cooled charge-coupled device (CCD) camera (400B eXcelon, Teledyn Princeton Instrument, United States) at a spectral resolution of $2\,cm^{-1}$ for spectroscopic measurements. The microfluidic chamber was connected to a passive pressure-driven laminar flow system, where the flows in each channel can be independently switched on/off through the fluidic valve. Prior to the measurements, AgNP-coated bead assemblies were flowed into the microfluidic chamber for overnight deposition. Next, the analyte solution, the buffer solution, and the AgNP solution were flowed into the microfluidic chamber in adjacent channels under laminar flow, which was maintained by adjusting the fluid flow rates in each channel to be equivalent. When the flow was ceased, analyte molecules and AgNPs were allowed to diffuse freely throughout the microfluidic chamber. The concentration of the AgNP solution was diluted to $10^{-11}$ M to avoid forming aggregates. A very few amounts of AgNP would diffuse into the adjacent sample channel for optical plasmonic trapping, thus the chance of trapping multiple AgNP simultaneously was kept at minimum. During the measurements, the power of 532 nm excitation laser was 2.7 mW and the power of 1064 nm trapping laser was 8 mW. Under the microscopic visualization, the microfluidic chamber was moved by a piezo-controlled micro-stage to place the junction of a deposited AgNP-coated bead dimer at the focus of the laser beams. By switching the 1064 nm trapping laser between on and off states, the freely diffusing AgNP was trapped and released at the plasmonic junction, forming the dynamic nanocavity among the trapped AgNP and the coated AgNP for high-throughput SERS measurements. The SERS spectra were collected at the junction of the AgNP-coated bead dimer in sample solutions and the background spectra were acquired in reference solutions. To adjust the pH condition during SERS measurements, 2 M NaOH solution was added through a side channel into the microfluidic chamber originally filled with 13 µL Tyr solution at pH 1.0. The pH was measured at the outlet of the microfluidic chamber by pH indicator strips and verified by stoichiometric calculations. For example, upon the addition of 1.5 µL of 2 M NaOH solution, the pH of the Tyr solution in the microfluidic chamber was anticipated to change from 1.0 to 10.5. A pH indicator

strip was placed right at the outlet, giving the specific colors in three reaction zones on the pH indicator strip that matched pH 10.5 with an accuracy of 0.5 pH unit. To characterize the structures of hIAPP, 10000 SERS spectra were collected at each early timepoint during the incubations under different pH conditions in multiple parallel experimental sessions for statistical analysis. All presented spectra were obtained upon the subtraction of the background and smoothed by Savitzky–Golay filter.

### CD characterization

The hIAPP solution with a concentration of 10 µM was extracted from the incubation at different time points and added to a quartz cuvette with a path length of 1 cm. Spectral measurements were conducted using a circular dichroism spectrometer (Chirascan V100, Applied Photophysics Ltd) under a room temperature of 25 °C. CD spectra were recorded at 1 nm intervals across the range from 200 to 260 nm, with a response time of 0.5 s. The acquired data were subjected to solvent background correction and presented in the unit of Molar Ellipticity [θ] (deg·cm²dmol⁻¹). Each CD spectrum is an average of 10 scans from 5 parallel measurements and smoothed by the Savitsky-Golay algorithm integrated into the spectrometer software.

### AFM measurement

A small aliquot (10 µL) of the hIAPP solution extracted from the incubation at different time point was deposited onto a freshly cleaved mica surface (0.5 cm × 0.5 cm) and allowed to adsorb for 30 minutes. The mica surface was subsequently rinsed with milli Q water and dried by compressed air. Surface scanning measurements were performed by the atomic force microscope in ScanAsyst mode (Dimension Icon, Bruker), using silicon cantilevers with a resonance frequency of ~300 kHz and a spring constant of ~40 N/m. Height and amplitude images were collected in air under a room temperature of 25 °C and a humidity of 50% at a scan rate of 1 Hz and a resolution of 512 × 512 pixels.

### Molecular dynamic (MD) simulation

MD simulations were performed using the GROMACS 2022 software package and AMBER 93 force field. The initial conformation of hIAPP monomer was taken from Hansmann and co-workers' unstructured hIAPP model[55], and the charge states were modified according to the pH and the $pK_a$ values of residues. The initial structure was centered in a cubic box with a distance of 1.0 nm to the box edge and solvated with simple point charge (SPC) water molecules. The system was neutralized with $Na^+$ and $Cl^-$ ions. Energy minimization was performed using the steepest descent algorithm, then the system was pre-equilibrated by the position-restrained simulation under canonical ensemble (NVT) and isothermal isobaric ensemble (NPT) in 200 ps. The production MD simulations were carried out a temperature of 310 K using the temperature coupling algorithm. The pressure was maintained at 1 atm using the Parrinello-Rahman barostat. The Particle Mesh Ewald (PME) method was used to calculate long-range electrostatic interactions, and the van der Waals interactions were truncated at a distance of 1.0 nm using the Verlet cutoff scheme. The LINCS algorithm was used to constrain the bond lengths, and all simulations were performed with periodic boundary conditions. The trajectory was saved every 2 ps for further analysis. The initial and final configurations along with the main conformational types observed in the representative simulation trajectories at pH 5.5 and pH 7.4 are illustrated in Supplementary Fig. 24 and Supplementary Fig. 25, respectively. Analysis of the MD simulations was performed using the GROMACS tools. The root-mean-square deviation (RMSD)-based cluster analysis was performed using the gmx cluster command with a Cα RMSD cutoff of 0.4 nm. Only the structures after 40 ns were used for the conformational analysis to avoid potential bias from the initial state. The definition of the Secondary

Structure of Proteins (DSSP) algorithm was used to analyze the secondary structure dynamics during the trajectory. In addition to the secondary structure evolutions of hIAPP shown in Figs. 4i and 5i, two supplementary short videos (Supplementary Movies 1 and 5) are generated by animating the trajectory frames from MD simulations. The representative conformations of hIAPP along the simulation trajectories shown in Figs. 4j and 5j, including initial conformations, predominant conformations, minor conformations, and final conformations, are further illustrated from x-y, y-z, and x-z plane perspectives in Supplementary Figs. 24 and 25, together with the short videos demonstrating the 360-degree rotations of these conformations in Supplementary Information (Supplementary Movie 2–4 and 6–10). The PDB files encompassing all conformations of hIAPP extracted from MD simulation trajectories are provided in Supplementary Information (Supplementary Data 1 and 2).

### Reporting summary

Further information on research design is available in the Nature Portfolio Reporting Summary linked to this article.

## Data availability

All data generated in this study are presented in the main text and the supplementary information. Source Data file has been deposited in Figshare under accession code: https://doi.org/10.6084/m9.figshare.23591901[67]. Source data are provided with this paper.

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

## Acknowledgements

We express our gratitude to Andrew D. Chesney and Prof. Ulrich H.E. Hansmann from the Department of Chemistry and Biochemistry at the University of Oklahoma for sharing hIAPP models and computational results, providing professional guidance on MD simulations, and engaging in insightful discussions on hIAPP amyloid aggregation. J.H. acknowledges the funding support from the Research Grant Council of Hong Kong under Project 26303018, 16309919, and 16309721. X.D. acknowledges Prof. Chi-Ming Che and his funding support from "The Laboratory for Synthetic Chemistry and Chemical Biology" under the Health@InnoHK Program launched by the Innovation and Technology Commission, The Government of Hong Kong Special Administrative Region of the People's Republic of China.

## Author contributions

W.F., H.C., W.L., and J.H. designed the experiments. W.F., H.C., X.D., H.Z., and V.M. performed the spectroscopic experiments and conducted the spectral data analysis. H.C. fabricated and analyzed nanostructures. W.F. and X.D. optimized microfluidic device for spectroscopic measurements. W F., X.D., and H.Z. conducted parallel sample preparations and characterizations. W.F. and V.M. performed the computational simulations. W.L. and J.H. supervised the project. All authors participated in the result discussion and the manuscript preparation.

## Competing interests

The authors declare no competing interests.
