## [Peer Review File · Nature Communications]

Efficient optical plasmonic tweezer-controlled single-molecule
SERS characterization of pH-dependent amylin species in
aqueous milieusReviewer #1 (Remarks to the Author):

This paper presents a platform to perform single molecule SERS by means of controlled optical (plasmonic) trapping.

The idea is based on the use of dimers of silica microspheres decorated with Ag nanoparticles that can couple and trap a third Ag particle. In the gap between the metals a strong field can be generated with very high spatial confinement enabling single molecule detection.

In order to demonstrate the single molecule sensitivity the authors used the standard bi-analytes SERS with convincing results.

then, the platform is used to perform some studies on the effect of different condition in the raman spectrum of different molecules. in particular, the effect of pH values used in the solution is investigated in details for Tyrosin and hIAPP.

the idea to use nanoparticles dimers or nanogaps between metallic nanoparticles to perform plasmonic trapping and single molecule sers is not new and several papers reported (almost recently) examples of similar experiments.

Here I don't understand the choice of Ag instead of the more stable Au. The stability of the Ag nanoparticles are not investigated in the manuscript although it is well known that different aqueous solutions can impact on the properties of the metal.

The most interesting part of the manuscript, and in my opinion the only one with some novelty, regards the detailed analysis of the effect of pH on the small peptide (hIAPP) here measured.

I have some major comments on the manuscript:

1. Figure 3 illustrates the comparison between experimental and DFT calculated spectra. The difference is significant and not well discussed in the manuscript.
2. Fig 4 a, b and c illustrate the experimental CD spectra obtained in different pH conditions. To me the illustrated behaviour is not convincing and not clear. this second part of the manuscript is very interesting but here it sounds rather speculative although some experimental data are reported and discussed. it could be interesting to get a look with some simulations on the molecular dynamic under different pH values.
3. At line 373 the authors say "Furthermore, identical amide I band at 1674 cm^{-1} and amide III band at 1226 cm^{-1} were observed from the SERS spectra of hIAPP incubated at pH 7.4 for the first 2 h then at pH 5.5 for the remaining 22 h." could they explain this better?

Therefore, the novelty in this paper is related to the section on the peptides analyses with pH, this could contribute to the field of research, but the analyses and the results need to be better discussed and performed. The support from some additional simulations (in particular molecular dynamics) could be a significant plus

Reviewer #2 (Remarks to the Author):

In the present manuscript, the authors develop a SERS methodology to study biomolecules in the aqueous environment.

Although the methodology is described well in the manuscript, we still need to make many adjustments. The investigation part of IAPP is incomplete, but the experimental method lacks essential details for reproducibility experiments in other laboratories. For example, the methodology of protein preparation is not reported. Same for AFM measurements etc.

The core of the problem needs to be better described in the introduction. A better description of the propriety of IAPP and its structure and a more thorough discussion of the properties of oligomers since the toxic oligomer hypothesis is the most accepted in the scientific community and oligomers are transient species, therefore difficult to characterize. This information can be found in Chem Rev 2021, 121, 1845-1893. Although in the abstract and some parts of the manuscript, the word "crowded" or synonyms appears, this word is incorrect in the context of these systems. It is correct to use an aqueous environment because "crowded" is a system that contains other biological species such as proteins, membranes, or high ionic strength.

Line 105: What does consistent flow rate mean? Maybe laminar flow. Discuss better.

Line 164: Quantify the noise because a signal (peak) can be considered as such if and only if its intensity exceeds three times the standard deviation in the noise.

Line 204: I don't think a single amino acid can dictate the structure-function relationship of a protein; otherwise, the folding problem would be solved.

Line 209: How was the pH measured? Specify.

Line 273: The concentration of 100 micro molar is too large since the behavior of IAPP strongly depends on concentration (10.1021/jp511758w). Also, why do the authors use a 10 micromolar solution of protein in SERS measurements and in CD measurements 100 micromolar? CD measurements must be repeated at the concentration used for SERS measurements.

Lines 313-316: Why do the authors not report the band of cysteines 2 and 7 that are characteristic of IAPP?

The authors discuss unlikely oligomeric species containing beta sheets. These beta-sheet-containing species have already been highlighted by molecular dynamics simulations and 2D-IR (10.1073/pnas.1314481110 ;10.1007/s00249-020-01424-1 , 10.1016/j.bbadis.2018.11.021). This needs to be specified in the manuscript. In addition, the authors performed multiple SERS measurements on the same or several independent samples. This is very important to establish, mainly if single-molecule experimental techniques are used to have a statistically significant sampling.

We thank the reviewers for their insightful critiques and helpful suggestions, and we have made the following changes in response to their valuable comments:

Reviewer #1 (Remarks to the Author):

This paper presents a platform to perform single molecule SERS by means of controlled optical (plasmonic) trapping.

The idea is based on the use of dimers of silica microspheres decorated with Ag nanoparticles that can couple and trap a third Ag particle. in the gap between the metals a strong field can be generated with very high spatial confinement enabling single molecule detection.

In order to demonstrate the single molecule sensitivity the authors used the standard bi-analytes SERS with convincing results.

then, the platform is used to perform some studies on the effect of different condition in the raman spectrum of different molecules. in particular, the effect of pH values used in the solution is investigated in details for Tyrosin and hIAPP.

the idea to use nanoparticles dimers or nanogaps between metallic nanoparticles to perform plasmonic trapping and single molecule sers is not new and several papers reported (almost recently) examples of similar experiments.

Here I don't understand the choice of Ag instead of the more stable Au. The stability of the Ag nanoparticles are not investigated in the manuscript although it is well known that different aqueous solutions can impact on the properties of the metal.

The most interesting part of the manuscript, and in my opinion the only one with some novelty, regards the detailed analysis of the effect of pH on the small peptide (hIAPP) here measured.

We greatly appreciate these thoughtful and constructive feedbacks. We have revised the entire manuscript to improve the logic flow and strengthen the novelty. In particular, we highlight the unique advantages of our platform (i.e. accessibility, efficiency, and high-throughput sampling capacity) by comparing it with other similar experiments. Although Au is more stable than Ag, we rationalize the choice of Ag by considering the enhancement factor and adding the stability assessments under different conditions. More importantly, we devote a more comprehensive investigation to elucidate the effect of pH on hIAPP by conducting complementary experiments in combined with molecular dynamics (MD) simulations.

We compare our platform to recent studies on the integration of optical plasmonic trapping and single molecule SERS detection with update citations in the introduction. (*Nat. Commun.* 2022; *J. Am. Chem. Soc.* 2022; *J. Phys. Chem. Lett.* 2021; etc.) Most of these platforms are fabricated by top-down methods, such as lithography techniques, which require specific instruments. As a new bottom-up approach, our platform is easy to make, adjust, and use with low cost and high efficiency. Owing to the novel framework of Ag nanoparticle-decorated silica microbead

dimer, the plasmonic junction can be visualized and located by a simple optical microscope. The interparticle gap at the junction can be adjusted using the programmable oligonucleotide linkers at a desired length. Moreover, it is the first time to utilize dynamic nanocavity formed by the on-and-off trapping of the third Ag nanoparticle at the plasmonic junction to enlarge both the SERS enhancement and the high-throughput sampling capacity for efficient single-molecule SERS detections. Hence, we introduce new strategies and accessible improvements to develop a practical single-molecule SERS platform, which enables us to study the effect of pH on tyrosine and hIAPP with statistically significant amounts of data in the applications.

Revised main text to highlight the unique advantages of our platform:

In comparison with existing optical plasmonic trapping experiments^{13,22–24,26–28,32,36}, this platform is easy to fabricate and locate without relying on expensive instruments, empowering high-throughput single-molecule SERS detections of freely diffusing samples in aqueous conditions.

We understand that Au nanoparticles are usually more stable than Ag nanoparticles, whereas Ag nanoparticles generally provide higher SERS enhancements than Au nanoparticles under comparable conditions. Previous studies reported that 50 nm Ag nanoparticles at 2.7×10^{-11} M were considerably stable at acidic, neutral, or basic pH over 24 h while increasing particle size could improve the colloidal stability against aggregation (*Int. J. Nanomedicine* 16, 3021-3040 (2021)). Thus, it is feasible to use the freshly prepared 70 nm Ag nanoparticles at 10^{-11} M within 0.5 h in our parallel measurement sessions.

Following the reviewer's reminder, we add supplementary experiments to prove the adequate stability of 70 nm Ag nanoparticle solutions with the concentration of 10^{-11} M at different pH. Specifically, we utilize dynamic light scattering (DLS) and zeta-potential measurements to monitor the size distributions and the surface properties of Ag nanoparticles over time. Supplementary Fig. 21 shows that the average hydrodynamic diameters maintained at around 70 nm and the zeta-potential values persisted below -25 mV over 2 h, indicating their stability against aggregation and oxidation. These Ag nanoparticles contain negative charges to exert electrostatic repulsion and stabilize colloidal dispersion. At pH 1.0, their stability was gradually decreased with the uptrend of the average hydrodynamic diameters and the zeta-potential values after 2 h, owing to the reduced electrostatic repulsion. To avoid potential interference, all Ag nanoparticles are newly prepared and immediately used at 10^{-11} M within 0.5 hour in parallel measurement sessions. Besides, the consistent optical properties and SERS activities of these Ag nanoparticles further confirms their stability. Hence, we leverage the short-term stability and the strong SERS enhancement of Ag nanoparticles to develop a dynamic single-

molecule SERS platform here. We will explore Au nanoparticles and Au-Ag bimetallic nanoparticles with the better long-term stability for other applications in future studies.

Revised method section to describe the stability test of Ag nanoparticles:

Colloidal AgNP solutions were prepared with the concentration of 10^{-11} M at different pH, and their stabilities were assessed by dynamic light scattering (DLS) and zeta-potential measurements (Supplementary Fig. 21). To ensure the constant average hydrodynamic diameters and zeta-potential values reflecting the stability of AgNP, all colloidal AgNP solutions were newly prepared and immediately used within 0.5 hour in parallel measurement sessions.

Newly added figure to show the results of the stability test of Ag nanoparticles:

Supplementary Fig. 21. *Plots of the time evolution of the average hydrodynamic diameters and the average zeta-potential of 10^{-11} M AgNP under different pH conditions. Data are shown as mean with errors derived from three parallel measurements.*

Of greater significance, we perform complementary experiments in conjunction with molecular dynamics (MD) simulations to elucidate the heterogeneous structures of hIAPP under the impact of two physiological pH, based on the reviewer's suggestions. The revised main text provides a more thorough analysis with novel molecular insights into hIAPP aggregation at early stages, which also offers a new strategy to study complex systems using an efficient single-molecule SERS platform. Below, we explain the detailed revisions in a point-by-point format.

I have some major comments on the manuscript:

1. Figure 3 illustrates the comparison between experimental and DFT calculated spectra. The difference is significant and not well discussed in the manuscript.

We improve the agreement between experimental and DFT calculated spectra and discuss the difference between them, which verifies the peak assignment and strengthens the interpretation of tyrosine's peak position and intensity changes under different pH environments.

Old Fig 3c directly compares the experimental spectra and the original DFT calculated spectra without adjusting any scaling factors. Although the downshift of the tyrosine ν_{8a} mode from the +1 charged state to the -2 charged state is consistent with the change in the experimental spectra, the vibrational frequencies in the higher wavenumber region (1600-1700 cm^{-1}) of the DFT calculated spectra deviate from the experiments by approximately 2-3%. It is common to observe the calculated frequencies higher than the experimental values especially in the higher wavenumber region, due to the systematic electronic structure insufficiencies, such as basis set deficiencies, electron-correlation effects, and nuclear motion treatment inaccuracies, which can be corrected by scaling factors. Hence, to compensate for these calculation errors, we apply an empirical scaling factor (0.9770) to the calculated frequencies in the wavenumber region above 1000 cm^{-1} as recommended in the literature. (*J. Comput. Chem.* 33, 2380–2387 (2012); *Appl. Spectrosc.* 63, 733–741 (2009); *J. Phys. Chem.* 99, 3093–3100 (1995)) In new Fig 3c, the DFT calculated frequencies agree with the experimental results better, giving the relative deviations less than 1%. At pH 1.0, the tyrosine doublet peaks are observed at 830 and 853 cm^{-1} and calculated at 826 and 858 cm^{-1} . The tyrosine ν_{8a} mode is observed at 1620 cm^{-1} and calculated at 1635 cm^{-1} . At pH 13.0, the tyrosine doublet peaks are observed at 836 and 854 cm^{-1} and calculated at 838 and 854 cm^{-1} . The tyrosine ν_{8a} mode is observed at 1602 cm^{-1} and calculated at 1592 cm^{-1} . Considering the approximations in calculations and the condition complexity in experiments, the spectral differences within a few wavenumbers are reasonable. Furthermore, the changes of peak positions and intensities in the DFT calculated spectra are in line with those in experimental spectra, providing a reliable support to the structural changes of tyrosine from the +1 charged state to the -2 charged state at different pH environments.

Revised main text on the DFT calculated spectra:

To understand the chemical environment of Tyr in aqueous solutions, we compared the SERS spectra of Tyr with the calculated Raman spectra of Tyr at different charge states in Fig. 3c. Using density functional theory (DFT) simulations, the construction of Tyr-H₂O clusters was based on Ghomi and co-workers' model by involving seven water molecules in the vicinity to coordinate with the amino group (pKa=9.2), the carboxyl group (pKa=2.2), and the phenol hydroxyl group (pKa=10.5) of Tyr (Supplementary Note. 2)⁴⁵, then their structures were further optimized at +1 and -2 charged states to match the experimental results at pH 1.0 and pH 13.0, respectively. Due to the inherent deficiencies related to DFT simulations, the calculated frequencies were overestimated especially in higher wavenumber region^{46,47}. An empirical scaling factor 0.9770 was applied to the wavenumbers above 1000 cm^{-1} in the simulated Raman spectra to align with the experimental observations.

New Fig 3c comparing the experimental spectra and the DFT calculated spectra of tyrosine:

Figure 3. pH-dependent structural transitions of Tyr. ... c Comparison of SERS spectra (upper panel) and simulated Raman spectra (lower panel) of Tyr at pH 1.0 (black) and pH 13.0 (red), respectively. The SERS spectra are adopted from (a). The simulated Raman spectra and electrostatic potential are generated from DFT calculations on Tyr in +1 charged state as $\text{NH}_3^+\text{CH}(\text{CH}_2\text{C}_6\text{H}_4\text{OH})\text{COOH}$ (d) and -2 charged state as $\text{NH}_2\text{CH}(\text{CH}_2\text{C}_6\text{H}_4\text{O}^-)\text{COO}^-$ (e), respectively. A scaling factor of 0.9770 was applied to the wavenumbers above 1000 cm^{-1} in simulated Raman spectra.

Old Fig 3c comparing the experimental spectra and the DFT calculated spectra of tyrosine:

Fig. 3. pH-dependent structural transitions of Tyr. ... c, Comparison of SERS spectra (upper panel) and simulated Raman spectra (lower panel) of Tyr at pH 1 (red) and pH 13 (black), respectively. The SERS spectra are adopted from (a). The simulated Raman spectra and

electrostatic potential are generated from DFT calculations on Tyr in +1 charged state as $\text{NH}_3^+\text{CH}(\text{CH}_2\text{C}_6\text{H}_4\text{OH})\text{COOH}$ (d) and -2 charged state as $\text{NH}_2\text{CH}(\text{CH}_2\text{C}_6\text{H}_4\text{O}^-)\text{COO}^-$ (e), respectively.

2. Fig 4 a, b and c illustrate the experimental CD spectra obtained in different pH conditions. To me the illustrated behaviour is not convincing and not clear. this second part of the manuscript is very interesting but here it sounds rather speculative although some experimental data are reported and discussed. it could be interesting to get a look with some simulations on the molecular dynamic under different pH values.

To better interpret the molecular behaviors of hIAPP under the influence of pH, we add molecular dynamics (MD) simulations of hIAPP at different pH conditions to depict its structural details for both ensembles and individuals. Moreover, we repeat the CD and SERS measurements of hIAPP at the same condition, reflecting the structural features of hIAPP at the bulk and at the single-molecule level, respectively, which serve as the experimental foundation for computational analysis. Good agreements between the results obtained from CD, SERS, and MD approaches facilitate a clear depiction of the hIAPP structures in different pH conditions and provide a stronger support to elucidate the effect of pH on its structural distribution and evolution associated with the complex amyloid aggregation processes.

Specifically, the CD spectra of hIAPP during the incubation at pH 5.5 present a peak at 203 nm with two shoulders at 212 and 223 nm, indicating the presences of random coil and α -helix in its ensemble conformations. These assignments are supported by the MD simulations of hIAPP monomers at +4 charged state under pH 5.5 condition, demonstrating the predominant coil and helix in the secondary structure evolution and content for its statistical assessment in the bulk. Moreover, it is consistent with the SERS spectra of hIAPP during the incubation at pH 5.5 with a statistically significant sampling size (10000 spectra at each timepoint) to reflect its ensemble structures. Statistics of the hIAPP structural features at the single-molecule level reconstructs its bulk structures and brings the unique information on population and probability of specific conformations, which enables a clearer comparison between SERS characterizations and MD simulations. Based on the consistency between these results, the predominating conformational snapshot of hIAPP during the incubation at pH 5.5 is identified, showing a mainly disordered structure at residues Lys-1 to Thr-4 and Thr-6 to Tyr-37 with a partial helix at residues Ala-5 to Leu-12.

Analogously, the CD spectra of hIAPP during pH 7.4 incubation at $t = 0$ h and $t = 2$ h show a peak at 203 nm with two shoulders at 212 and 223 nm to imply the ensemble conformation with predominant random coil and partial helix at the early stage of aggregation, followed by

a dramatic change into a strong peak at 218 nm at $t = 24$ h to suggest the formation of amyloid fibrils with β -sheet structure at the later stage of aggregation. Focusing on the early stage of aggregation, the MD simulations of hIAPP monomers at +3 charged state under pH 7.4 condition present the predominating coil and helix and the emerging turn and β -sheet fractions in the secondary structure evolution and content for its statistical assessment in the bulk, which verifies the assignments of CD spectra at $t = 0$ h and $t = 2$ h and explains the slight deviation between them. Moreover, these assignments are supported by the SERS spectra of hIAPP during the incubation at pH 7.4 with a statistically significant sampling size (10000 spectra at each timepoint) to reflect both its ensemble structures and individual conformations with the unique information on their population and probability. According to the good arrangements between experimental observations and MD simulations, three most-populated conformations are identified: The predominant conformation exhibits the partial helix-coil structure. The type I minor conformation has a turn at residues 20-25. The type II minor conformation contains a short β -hairpin structure (residue 14-19 and 26-29), giving lower values of the root-mean-square fluctuation (RMSF) at residues 30-37 than that of the type I conformer to imply a more constrained C-terminal. Significantly, these minor conformations in MD simulation could play an important role to slightly shift the ensemble structures of hIAPP in the CD spectra at $t = 2$ h and interpret the low-populated type I and type II SERS spectra of hIAPP at $t = 2$ h incubation.

Interestingly, after the incubation of hIAPP at pH 7.4 condition for the first 2 hours, we adjust the condition to pH 5.5 to continue the incubation for the additional 22 hours and obtain the CD spectrum with a strong peak at 218 nm at $t = 24$ h (total incubation time), indicating the formation of amyloid fibrils with β -sheet structure. That means the critical species might have formed during the first 2-hour incubation of hIAPP at pH 7.4 condition to drive the aggregation even though the incubation condition was adjusted to pH 5.5 for the following 22 hours. The early species of hIAPP at pH 7.4 holds significantly implications for the irreversible amyloid formation regardless of pH conditions in succeeding incubation. Due to the limited computing resource, we cannot perform long-time MD simulations. Fortunately, a computational group share with us that the given hIAPP conformations at pH 7.4 can fold into the ordered β -sheet structure similar to its amyloid fibril during the simulations over 3800 ns. (We acknowledge that this information regarding the long-time MD simulations of hIAPP is being shared as a courtesy and the full results will be reported separately by that computational group.) Overall, the pH-regulated hIAPP conformational evolution identified by our CD, SERS experiments and MD simulations are consistent with previous 2D-IR, SERS, and MD studies, linking these early species of hIAPP to the development of amyloid aggregation.

Revised Fig. 4 and 5 adding the hIAPP structural details obtained from MD simulations:

Figure 4. Structural characterizations of hIAPP incubated under pH 5.5. **a** Properties of hIAPP residues with +4 charges at pH 5.5: Positively charged residues in red, hydrophobic residues in blue, and residues with helical potency in the dashed box. **b** CD spectra of 10 μ M hIAPP solution under pH 5.5 incubation at $t = 0, 2,$ and 24 h. **c** Representative SERS spectra of 10 μ M hIAPP solution under pH 5.5 incubation at $t = 0$ h (black) and $t = 2$ h (green). **d** The disulfide bond region of (c). **e** The Tyr doublets region of (c). **f** The Amide I region of (c). **g** Secondary derivative spectra of (f). **h** Mapping of secondary derivative spectra of the Amide I region of the SERS spectra of hIAPP at $t = 2$ h incubation under pH 5.5 from parallel experimental sessions. The color bar shows the normalized intensities from low (dark blue) to high (red-black). **i** Representative time evolution of secondary structure per residue of monomeric hIAPP with +4 charges at pH 5.5. Color code: blue: helix; white: random coil; green: bend; yellow: turn; red: β -sheet. **j** Snapshots of the predominant conformation from (i). **k** Secondary structural contents of disordered, helix, turn, and β -sheet of the monomeric hIAPP from (i).

Figure 5. Structural characterizations of hIAPP incubated under pH 7.4. **a** Properties of hIAPP residues with +3 charges at pH 7.4: Positively charged residues in red, hydrophobic residues in blue, and His-18 highlighted in yellow as the deprotonated residue upon the change of pH from 5.5 to 7.4. **b** CD spectra of 10 μ M hIAPP solution under pH 7.4 incubation at $t = 0, 2,$ and 24 h. **c** Representative SERS spectra of 10 μ M hIAPP solution under pH 7.4 incubation at $t = 0$ h (black) and $t = 2$ h (most-populated spectrum, green; type I rare spectrum, purple; type II rare spectrum, red). **d** The disulfide bond region of (c). **e** The Tyr doublets region of (c). **f** The Amide I region of (c). **g** Secondary derivative spectra of (f). **h** Mapping of secondary derivative spectra of the Amide I region of the SERS spectra of hIAPP at $t = 2$ h incubation under pH 7.4 from parallel experimental sessions. The color bar shows the normalized intensities from low (dark blue) to high (red-black). **i** Representative time evolution of secondary structure per residue of monomeric hIAPP with +3 charges at pH 7.4. Color code: blue: helix; white: random coil; green: bend; yellow: turn; red: β -sheet. **j** Snapshots of the predominant conformation, the type I minor conformation, and the type II minor conformation from (i). **k** Secondary structural contents of disordered, helix, turn, and β -sheet of the monomeric hIAPP from (i).

Revised Supplementary Fig. 17 showing the CD spectra of hIAPP during incubation with the pH adjustment from pH 7.4 to 5.5:

Supplementary Fig. 17. e CD spectra of 10 μM hIAPP solution at $t = 0, 2,$ and 24 h in the incubation under pH 7.4 for the first 2 hours and then adjusted to pH 5.5 for the following 22 hours.

Revised main text on the discussion of CD spectra involving the newly added MD simulations: The CD spectra of 10 μM hIAPP in the PBS buffer of pH 5.5 for incubation at $t = 0, 2,$ and 24 h present a persistent negative peak at around 203 nm with two shoulders at 212 and 223 nm in Fig. 4b (black, green, blue), which are consistent with previous studies⁵¹ to confirm the ensemble conformation as major random coils and minor α -helix⁵.....Furthermore, MD simulations were conducted to monitor the monomeric conformational dynamics of hIAPP at +4 charged state under pH 5.5 condition, where the initial input was modified from Hansmann and co-workers' unstructured hIAPP model⁵³ to reflect experimental observations. In Fig. 4i and 4k, the secondary structure evolution and the statistics of structural contents show the extensive disordered structures with little helix and turn segments of hIAPP at pH 5.5, consistent with the CD and SERS spectra for its bulk assessments. Specifically, the predominating hIAPP conformation adopts mainly random coils with a short helix near N-terminal, as illustrated in Fig. 4j. The high conformational flexibility and electrostatic repulsions could give rise to the swollen coil conformation and plausible inhibitory effect on fibril development^{5,54}, as proven by the persisting predominant helix-coil structures in the CD and SERS of hIAPP after the 24-hour incubation at pH 5.5.

In Fig. 5b, the CD spectra of 10 μM hIAPP under pH 7.4 incubation at $t = 0$ h and $t = 2$ h (black and green) show a peak at 203 nm with two shoulders at 212 and 223 nm to imply the ensemble conformation with major random coil and minor helix structures at the early stage of aggregation, followed by a dramatic change into a strong peak at 218 nm at $t = 24$ h (blue) to suggest the formation of amyloid fibrils with β -sheet structure at the later stage of

aggregation⁵.....MD simulations of the monomeric hIAPP at pH 7.4 demonstrate the emerging turn and β -sheet fractions among the predominating helix and coil structures in the secondary structure evolution and content in Fig. 5i and 5k. Top three most-populated conformations are illustrated in Fig. 5j. The predominant conformation exhibits the partial helix-coil structure. The type I minor conformation has a turn at residues 20-25. The type II minor conformation contains a short β -hairpin structure (residue 14-19 and 26-29), giving lower values of the root-mean-square fluctuation (RMSF) at residues 30-37 than that of the type I conformer to reflect a more constrained C-terminal (Supplementary Fig. 16). These results are in good agreement with the experimental observations and consistent with previous MD simulations⁵⁷⁻⁵⁹, providing molecular details of the monomeric species that might contribute to the type I and II spectra of hIAPP at the very early aggregation stage.

3. At line 373 the authors say "Furthermore, identical amide I band at 1674 cm^{-1} and amide III band at 1226 cm^{-1} were observed from the SERS spectra of hIAPP incubated at pH 7.4 for the first 2 h then at pH 5.5 for the remaining 22 h." could they explain this better?

To improve clarity, we revised the main text and the supplementary figures on the comparison between two parallel experiments: 1. The incubation of hIAPP at pH 7.4 for 24 hours, and 2. The incubation of hIAPP at pH 7.4 for the initial 2 hours, followed by the pH adjustment from 7.4 to 5.5 to continue the incubation for the following 22 hours. Both of them generated the amyloid fibrils of hIAPP at $t = 24$ h (total incubations time) as characterized by CD, AFM, and SERS measurements. Specifically, the SERS spectrum of the amyloid fibrils formed after the 24-hour incubation involving the pH adjustment from 7.4 to 5.5 exhibits the amide I band at 1672 cm^{-1} and amide III band at 1226 cm^{-1} , which are identical to the characteristic features of the typical hIAPP fibrils produced by the 24-hour incubation at pH 7.4. These results confirm the formation of hIAPP amyloid fibrils initiated by the 2-hour incubation at pH 7.4, regardless of the pH adjustment from 7.4 to 5.5 at $t = 2$ h for the subsequent 22-hour incubation at pH 5.5.

Revised main text on the incubation of hIAPP involving the adjustment of pH from 7.4 to 5.5: *For comparison, we incubated 10 μM hIAPP at pH 7.4 for an initial period of 2 hours, followed by a subsequent adjustment of pH from 7.4 to 5.5 to continue the incubation for the succeeding 22 hours.....We then utilized AgNP colloids to perform the bulk SERS measurement of these amyloid fibrils generated after the 24-hour incubation involved the pH adjustment from 7.4 to 5.5 at $t = 2$ h. Their SERS spectrum in Supplementary Fig. 19 exhibits the amide I band at 1674 cm^{-1} and amide III band at 1226 cm^{-1} indicative of the ordered β -sheet structure⁵⁰ and the peak at 490 cm^{-1} attributed to the strained disulfide bond⁵², which are identical to the characteristic features of typical hIAPP fibrils with compact β -sheet structures generated from the 24-hour incubation at pH 7.4 (Supplementary Fig. 20). These results confirm the formation of hIAPP amyloid fibrils despite the pH adjustment to 5.5 for the succeeding incubation.*

Previous main text near line 373 on the SERS spectrum of hIAPP amyloid fibrils:

Furthermore, identical amide I band at 1674 cm^{-1} and amide III band at 1226 cm^{-1} were observed from the SERS spectra spectrum of hIAPP incubated at pH 7.4 for the first 2 h then at pH 5.5 for the remaining 22 h.

Revised figures on the incubation of hIAPP at different pH conditions:

Supplementary Fig. 19. SERS spectrum of the amyloid fibrils generated after the 24-hour incubation of hIAPP with the initial incubation at pH 7.4 for 2 hours, followed by the adjustment of pH from 7.4 to 5.5 to continue the subsequent incubation at pH 5.5 for the succeeding 22 hours.

Supplementary Fig. 20. SERS spectrum of the amyloid fibrils generated after the 24-hour incubation of hIAPP at pH 7.4.

Supplementary Fig. 17. a CD spectra of 10 μM hIAPP solution under pH 5.5 incubation at $t = 0, 2,$ and 24 h. **b** The absolute intensities at 218 nm of the CD spectra as a function of time during the incubation of hIAPP under pH 5.5 for 24 hours. **c** CD spectra of 10 μM hIAPP solution under pH 7.4 incubation at $t = 0, 2,$ and 24 h. **d** The absolute intensities at 218 nm of the CD spectra as a function of time during the incubation of hIAPP under pH 7.4 for 24 hours. **e** CD spectra of 10 μM hIAPP solution at $t = 0, 2,$ and 24 h in the incubation under pH 7.4 for the first 2 hours and then adjusted to pH 5.5 for the following 22 hours. **f** The absolute intensities at 218 nm of the CD spectra as a function of time during the incubation of hIAPP under pH 7.4 for the first 2 hours, followed by the adjustment from pH 7.4 to 5.5 to continue the incubation for the succeeding 22 hours.

Therefore, the novelty in this paper is related to the section on the peptides analyses with pH, this could contribute to the field of research, but the analyses and the results need to be better discussed and performed. The support from some additional simulations (in particular molecular dynamics) could be a significant plus

We express our sincere gratitude for the reviewer's thoughtful insight and acknowledgement of the potential contribution of our paper to the research field. To strengthen the novelty and improve the quality, we add new experiments and MD simulations to analyze and discuss the molecular behaviors of hIAPP under the influence of pH, as suggested. The simulation results well match with the experimental observations, depicting the structural details of hIAPP under different pH conditions at both the bulk level and the single-molecule level to better elucidate the structural distribution and evolution of hIAPP during its early stages of aggregation, which substantially enhances the credibility and significance of our findings on the hIAPP transient species among its monomers as the critical intermediates towards irreversible fibrillation to advance the comprehension of amyloid aggregation associated with type II diabetes.

Reviewer #2 (Remarks to the Author):

In the present manuscript, the authors develop a SERS methodology to study biomolecules in the aqueous environment.

Although the methodology is described well in the manuscript, we still need to make many adjustments. The investigation part of IAPP is incomplete, but the experimental method lacks essential details for reproducibility experiments in other laboratories. For example, the methodology of protein preparation is not reported. Same for AFM measurements etc.

We are grateful for these insightful and thorough comments. We have substantially revised the manuscript to enhance both quality and clarity. Specifically, we devote a more comprehensive investigation on hIAPP at the early lag phase prior to amyloid formation by incorporating complementary experiments, molecular dynamics (MD) simulations, and relevant literatures in greater depth. Moreover, we add the detailed descriptions regarding the experimental methodology for protein preparation, AFM measurement, and CD characterization to ensure precision and reproducibility, as suggested.

Newly added method section on protein preparation:

hIAPP preparation

hIAPP containing the disulfide bridge between Cys-2 and Cys-7 and the amidated C-terminal was purchased from GL Biochem (Shanghai) Ltd. It was dissolved in hexafluoro-isopropanol (HFIP) at

a concentration of 1 mg/mL and incubated for 1 h, followed by lyophilization. Then, the purified hIAPP was rehydrated in 10 mM PBS buffer and filtered through a 0.22 nm Tuffryn syringe filter to prepare the stock solutions with a concentration of 1 mg/mL (250 μ M). In parallel experimental sessions, several batches of the hIAPP stock solutions were prepared as independent samples for the subsequent incubations at different pH conditions. Parts of the hIAPP stock solutions were diluted to the concentration of 10 μ M and adjusted to pH 5.5, followed by the 24-hour incubation at 37 °C. Parts of the hIAPP stock solutions were diluted to the concentration of 10 μ M and adjusted to pH 7.4, followed by the 24-hour incubation at 37 °C. Parts of the hIAPP stock solutions were diluted to the concentration of 10 μ M and adjusted to pH 7.4 for a 2-hour incubation at 37 °C, followed by the subsequent adjustment from pH 7.4 to 5.5 to continue the succeeding 22-hour incubation at 37 °C. Meanwhile, the time-dependent CD, SERS, and AFM characterizations were conducted repeatedly for parallel experimental sessions during the incubations of these hIAPP solutions from different batches of independent preparations.

Newly added method section on AFM measurement:

AFM measurement

A small aliquot (10 μ L) of the hIAPP solution extracted from the incubation at different time point was deposited onto a freshly cleaved mica surface (0.5 cm \times 0.5 cm) and allowed to adsorb for 30 minutes. The mica surface was subsequently rinsed with milli Q water and dried by compressed air. Surface scanning measurements were performed by the atomic force microscope in ScanAsyst mode (Dimension Icon, Bruker), using silicon cantilevers with a resonance frequency of \sim 300 kHz and a spring constant of \sim 40 N/m. Height and amplitude images were collected in air under the room temperature of 25 °C and the humidity of 50% at a scan rate of 1 Hz and a resolution of 512 \times 512 pixels.

Newly added method section on CD characterization:

CD characterization

The hIAPP solution with the concentration of 10 μ M was extracted from the incubation at different time point and added to a quartz cuvette with a path length of 1 cm. Spectral measurements were conducted using a circular dichroism spectrometer (Chirascan V100, Applied Photophysics Ltd) under the room temperature of 25 °C. CD spectra were recorded at 1 nm intervals across the range from 200 to 260 nm, with a response time of 0.5 s. The acquired data were subjected to solvent background correction and presented in the unit of Molar Ellipticity $[\theta]$ ($\text{deg}\cdot\text{cm}^2\cdot\text{dmol}^{-1}$). Each CD spectrum is an average of 10 scans from 5 parallel measurements and smoothed by Savitsky-Golay algorithm integrated in the spectrometer software.

The core of the problem needs to be better described in the introduction. A better description of the propriety of IAPP and its structure and a more thorough discussion of the properties of oligomers since the toxic oligomer hypothesis is the most accepted in the scientific community and oligomers are transient species, therefore difficult to characterize. This information can be found in Chem Rev 2021, 121, 1845-1893.

We revise the introduction in response to the reviewer's suggestions and add the appropriate citations to *Chem. Rev.* 2021, 121, 1845-1893 and other relating literatures. Specifically, hIAPP possesses intrinsic disordered structures with a high aggregation propensity governed by amino acid sequence (i.e. charge, hydrophobicity, and aromaticity) and surrounding environment (i.e. pH, insulin, ions, and lipids) to form amyloid fibrils with distinct β -sheet structures in most type II diabetes patients. The early-stage aggregation of hIAPP presents a complex process involving the formation of various oligomeric intermediates with different conformations and stabilities at low concentrations among the predominating monomers in a dynamic mixture. The crucial role of hIAPP oligomers in causing cytotoxicity and driving amyloid aggregation is widely recognized by the scientific community. However, due to the transient nature and structural heterogeneity of hIAPP oligomers, it is challenging to characterize their detailed structures and the conversion from monomers to oligomers in amyloid aggregation processes, calling for the development of new biophysical approaches to remove ensemble averaging to study different co-existing species in complex mixtures.

Revised introduction on the significance and challenge to characterize hIAPP oligomers:

For example, human Islet Amyloid Polypeptide (hIAPP) lacks stable secondary structures but possesses the aggregation propensity governed by its intrinsic sequence and surrounding environment to form amyloid fibrils with distinct β -sheet structures in type II diabetes patients⁴. The aggregation of hIAPP is repressed in the secretory granules of pancreatic β -cells at pH 5.5 and millimolar (mM) concentration while promoted in certain extracellular compartments at pH 7.4 and micromolar (μ M) concentration⁵, generating various oligomeric intermediates at low populations in a dynamic mixture towards fibril formation⁶. The crucial role of hIAPP oligomers in causing cytotoxicity and driving amyloid aggregation is widely recognized in literatures^{4,6,7}. However, owing to their transient nature and structural heterogeneity, it is challenging to characterize the detailed structures of hIAPP oligomers and the conversion from monomers to oligomers in complex amyloid assembly processes without excluding the influences of surrounding environment^{4,8}.

Previous introduction on hIAPP:

For example, human Islet Amyloid Polypeptide (hIAPP) is a typical IDP that lacks stable secondary structures in the secretory granules of pancreatic β -cells at around pH 5.5 and

millimolar concentration⁶. Whereas, hIAPP misfolds and assembles in certain extracellular compartments at pH 7.4 and micromolar concentration⁶, generating low-populated, heterogenous, and transient species in a dynamic equilibrium mixture to induce cytotoxicity and initiate the growth of amyloid fibrils^{7,8}, which are commonly found in type II diabetes patients⁹. However, the molecular mechanism of the amyloid formation, especially at the early stage, remains unclear since it is challenging to probe these rare species without excluding the influences of surrounding molecules and micro-environments¹⁰.

Although in the abstract and some parts of the manuscript, the word “crowded” or synonyms appears, this word is incorrect in the context of these systems. It is correct to use an aqueous environment because “crowded” is a system that contains other biological species such as proteins, membranes, or high ionic strength.

Thanks for pointing out this inaccurate word usage. We replace “crowded” and its synonyms with “aqueous” to precisely reflect the molecular environment, since we study hIAPP at the concentration of 10 μ M in the buffer solutions at different physiological pH, not the crowded conditions containing other biological species. The revisions are marked in the manuscript.

Line 105: What does consistent flow rate mean? Maybe laminar flow. Discuss better.

We appreciate the careful review in identifying this confusion. We revise “consistent flow” to “laminar flow” and describe the microfluidic system with more appropriate language. In our microfluidic chamber, fluids flow in three parallel layers without mixing, which is a typical laminar flow. The laminar flow is maintained by controlling the fluid flow rates in each channel to ensure that the fluid velocities are the same at the interface between channels. When the flow is suspended, free diffusion occurs throughout the microfluidic chamber.

Revised main text near line 105 (now 87) on the microfluidic system:

Prior to spectroscopic measurements, the analyte solution, the buffer solution, and the AgNP solution were injected into the three adjacent channels in the microfluidic chamber under laminar flow, which was maintained by adjusting the fluid flow rates in each channel to be equivalent. Upon cessation of the flow, analyte molecules and AgNPs were allowed to diffuse freely throughout the microfluidic chamber.

Previous main text near line 105 on the microfluidic system:

Before the spectroscopic measurements, the solutions of analyte molecules and AgNP were injected into the microfluidic chamber in adjacent channels under a consistent flow rate.

Line 164: Quantify the noise because a signal (peak) can be considered as such if and only if its intensity exceeds three times the standard deviation in the noise.

We adhered to this convention in spectral assessments, but not presented it clearly. Following the reviewer's suggestion, we include a quantitative analysis of the noise (σ) and demonstrate the intensity cutoff criterion for distinguishing signals from background noise at three times the standard deviation in the noise (3σ).

Newly added figure to show the noise analysis in SERS measurements:

Supplementary Fig. 7. **a**, Original background spectrum and noise analysis, giving the standard deviation 1σ as 20 counts (-1σ as -20 counts). **b**, Representative spectrum of "single-MB" event. **c**, Representative spectrum of "single-NBA" event. **d**, Representative spectrum of "Dual-MB and NBA" event, showing the intensity cutoff criterion for distinguishing signals from background noise at 3σ as 60 counts.

Revised main text regarding the spectral assessment:

Statistical analysis of the data indicates that ~5 % spectra exhibited peaks intensities above three times the standard deviation of noise (Supplementary Fig. 7) and classified into the catalog of 'single-MB event', 'single-NBA event', or 'dual-MB and NBA event',

Revised main text near line 164 (now 138) on the SERS mapping figure:

Only the spectra with the peak intensities above three times the standard deviation of the noise are shown.

Previous main text near line 164 on the SERS mapping figure:

Only the spectra above the noise level were shown.

Line 204: I don't think a single amino acid can dictate the structure-function relationship of a protein; otherwise, the folding problem would be solved.

We agree that the structure-function relationship of a protein should be dictated by more than a single amino acid. To improve the clarity, we revise the wording and phrasing to explain that the properties of amino acid side chains could ultimately affect the folding of proteins through a variety of interactions between numerous constituent amino acids.

Revised main text near line 204 (now 171) on the background of amino acid tyrosine (Tyr):

To further verify the aqueous stability and compatibility, we employed the optical plasmonic tweezer-coupled SERS platform to characterize tyrosine (Tyr) in different pH environments, considering that the side chains of numerous constituent amino acids interact to facilitate protein folding while their hydrogen bonding and electrostatic interactions are affected by environmental factors, such as pH³⁹.

Previous main text near line 204 on the background of amino acid tyrosine (Tyr):

To further verify the aqueous stability and compatibility, we employed the optical plasmonic tweezer-coupled SERS platform to characterize tyrosine (Tyr) in different pH environments, considering that the side chains of amino acids dictate the protein structure-function relationships while their hydrogen bonding and electrostatic interactions are affected by environmental factors, such as pH⁴¹.

Line 209: How was the pH measured? Specify.

We elaborate the pH measurement in method section and revise the main text. Specifically, the pH was measured by pH indicator strips at the outlet of the microfluidic chamber and verified

by calculating the amount of 2 M NaOH solution added to the microfluidic chamber originally filled with 13 μL Tyr solution at pH 1.0. For example, upon the addition of 1.5 μL of 2 M NaOH solution, the pH of the Tyr solution in the microfluidic chamber is anticipated to change from 1.0 to 10.5. A pH indicator strip is placed right at the outlet, giving the specific colors in three reaction zones on the pH indicator strip that matches pH 10.5 with an accuracy of 0.5 pH unit.

Newly added method section to elaborate the pH adjustment and measurement:

To adjust the pH condition during SERS measurements, 2M NaOH solution was added through a side channel into the microfluidic chamber originally filled with 13 μL Tyr solution at pH 1.0. The pH was measured at the outlet of the microfluidic chamber by pH indicator strips and verified by stoichiometric calculations. For example, upon the addition of 1.5 μL of 2 M NaOH solution, the pH of the Tyr solution in the microfluidic chamber was anticipated to change from 1.0 to 10.5. A pH indicator strip was placed right at the outlet, giving the specific colors in three reaction zones on the pH indicator strip that matched pH 10.5 with an accuracy of 0.5 pH unit.

Revised main text near line 209 (now 177) on the pH adjustment and measurement:

Then 2 M NaOH was added through a side channel to adjust the environmental pH from 1.0 to 13.0, which was measured at the outlet of the microfluidic chamber by pH indicator strips and verified by stoichiometric calculations. Meanwhile, the corresponding SERS spectra were collected continuously.

Previous main text near line 209 on the pH adjustment:

Then 2 M NaOH was added through a side channel to adjust the environments from pH 1 to pH 13, meanwhile SERS spectra were continuously collected.

Line 273: The concentration of 100 micro molar is too large since the behavior of IAPP strongly depends on concentration (10.1021/jp511758w). Also, why do the authors use a 10 micromolar solution of protein in SERS measurements and in CD measurements 100 micromolar? CD measurements must be repeated at the concentration used for SERS measurements.

We thank the reviewer for providing the helpful reference and suggestion to improve the rigor of this study. As suggested, we repeat the CD measurements of 10 μM IAPP solution at several time points during incubation, referring to the change of kinetics and mechanism of IAPP at the critical concentrations in the low micromolar range and other factors reported in the paper (10.1021/jp511758w). Consistency has been achieved in terms of the IAPP concentrations in CD measurements and SERS measurements in the revised manuscript with this citation.

Revised main text near line 273 (now 252) on the CD measurements:

The CD spectra of 10 μM hIAPP in the PBS buffer of pH 5.5 for incubation at $t = 0, 2,$ and 24 h present a persistent negative peak at around 203 nm with two shoulders at 212 and 223 nm in Fig. 4b (black, green, blue), which are consistent with previous studies⁵¹ to confirm the ensemble conformation as major random coils and minor α -helix⁵.

Previous main text near line 273 on the CD measurements:

The CD spectra of 100 μM hIAPP in the PBS buffer of pH 5.5 at different incubation times (0 h, 2 h, 24 h) present a persistent negative peak at around 203 nm in Fig. 4a (black, green, blue), indicating that the ensemble conformation is dominated by random coils⁶.

Newly added method section on CD measurements:

CD characterization

The hIAPP solution with the concentration of 10 μM was extracted from the incubation at different time point and added to a quartz cuvette with a path length of 1 cm. Spectral measurements were conducted using a circular dichroism spectrometer (Chirascan V100, Applied Photophysics Ltd) under the room temperature of 25 $^{\circ}\text{C}$. CD spectra were recorded at 1 nm intervals across the range from 200 to 260 nm, with a response time of 0.5 s. The acquired data were subjected to solvent background correction and presented in the unit of Molar Ellipticity $[\theta]$ ($\text{deg}\cdot\text{cm}^2\cdot\text{dmol}^{-1}$). Each CD spectrum is an average of 10 scans from 5 parallel measurements and smoothed by Savitsky-Golay algorithm integrated in the spectrometer software.

Revised figures showing the new CD spectra of 10 μM hIAPP during different incubations:

Figure 4. b CD spectra of 10 μM hIAPP solution under pH 5.5 incubation at $t = 0, 2,$ and 24 h.

Figure 5. b CD spectra of 10 μM hIAPP solution under pH 7.4 incubation at $t = 0, 2,$ and 24 h.

Supplementary Fig. 17. e CD spectra of 10 μM hIAPP solution at $t = 0, 2,$ and 24 h in the incubation under pH 7.4 for the first 2 hours and then adjusted to pH 5.5 for the following 22 hours.

Lines 313-316: Why do the authors not report the band of cysteines 2 and 7 that are characteristic of IAPP?

In response to the reviewer's reminder, we repeat the SERS measurements and include the region of 400-600 cm^{-1} to characterize the disulfide bond between Cys2-Cys7 of hIAPP at the concentration of 10 μM during incubation under different pH conditions. As shown in the new SERS spectra below, a distinct peak at 523 cm^{-1} is consistently observed before and after the incubation of 10 μM hIAPP solution at pH 7.4, indicating the disulfide bridge between Cys2 and Cys7 with a gauche-gauche-trans/trans-gauche-gauche (g-g-t/g-g) conformation. A peak at 490 cm^{-1} emerges at the late stage of hIAPP incubation, which is attributed to the strained

disulfide bond with a low dihedral angle, reflecting mechanical stress imposed on the disulfide bond due to the formation of amyloid fibrils. The absence of this peak before incubation implies that hIAPP exists primarily in monomeric or low-order oligomeric forms without forming compact aggregates or amyloid fibrils (10.1016/j.jsb.2017.06.002). These spectral features can serve as reliable indicators to characterize the conformations of hIAPP at the early aggregation stages under the influence of pH conditions, as highlighted in the revised main text and figures.

Newly added figures showing the bands of cysteines 2 and 7 of hIAPP:

Supplementary Fig. 15. a Representative SERS spectra of hIAPP before the incubation at the concentration of 10 μM under pH 7.4 condition. **b** Representative SERS spectra of hIAPP after the incubation at the concentration of 10 μM under pH 7.4 condition for 24 hours, showing the disulfide bond between Cys2-Cys7 of hIAPP in the region of 400-600 cm^{-1} .

Revised main text near line 313-316 (now 262-265) on the peak assignments of hIAPP:

The characteristic peaks are assigned to protein backbones including amide I band (1656 cm^{-1}), CH₂ deformation (1450 cm^{-1}) and amide III band (1250 cm^{-1} and 1287 cm^{-1}), as well as specific residues such as Phe (1006 cm^{-1} , 1585 cm^{-1}), Tyr (830 cm^{-1} and 850 cm^{-1} , 1605 cm^{-1}), and Cys-Cys (523 cm^{-1})⁵².

Previous main text near line 313-316 on the peak assignments of hIAPP:

The characteristic peaks are assigned to protein backbones including amide I band (1656 cm^{-1}), CH₂ deformation (1450 cm^{-1}) and amide III band (1250 cm^{-1} and 1287 cm^{-1}), as well as specific residues such as Phe (1006 cm^{-1} , 1585 cm^{-1}) and Tyr (830 cm^{-1} and 850 cm^{-1} , 1605 cm^{-1})^{58,59}.

Newly added main text discussing the conformations of hIAPP at different pH conditions:

In addition, the peak at 523 cm⁻¹ represents the gauche-gauche-trans/trans-gauche-gauche (g-g-t/t-g-g) conformation of the disulfide bridge between Cys2 and Cys7⁵². Notably, the spectra of type I and type II species both contain the peak at 523 cm⁻¹, indicative of the intact disulfide bond to stabilize the N-terminal helix structure. There is no obvious peak at 490 cm⁻¹, which is related to disulfide bond restrictions (Supplementary Fig. 15), implying these hIAPP species in primary monomeric or low-order oligomeric forms without forming compact aggregates or amyloid fibrils⁵².

Newly added Fig. 4 **c** and **d** showing the bands of cysteines 2 and 7 of hIAPP under pH 5.5:

Figure 4. **c** Representative SERS spectra of 10 μM hIAPP solution under pH 5.5 incubation at $t = 0$ h (black) and $t = 2$ h (green). **d** The disulfide bond region of (c).

Newly added Fig. 5 **c** and **d** showing the bands of cysteines 2 and 7 of hIAPP under pH 7.4:

Figure 5. **c** Representative SERS spectra of 10 μM hIAPP solution under pH 7.4 incubation at $t = 0$ h (black) and $t = 2$ h (most-populated spectrum, green; type I rare spectrum, purple; type II rare spectrum, red). **d** The disulfide bond region of (c).

The authors discuss unlikely oligomeric species containing beta sheets. These beta-sheet-containing species have already been highlighted by molecular dynamics simulations and 2D-IR (10.1073/pnas.1314481110 ;10.1007/s00249-020-01424-1, 10.1016/j.bbadis.2018.11.021). This needs to be specified in the manuscript.

We appreciate the reviewer for providing the relevant literatures to support our experimental observations. To strengthen the scientific depth of our work, we add a detailed discussion with reference to the previous studies on these beta-sheet-containing hIAPP species. Moreover, we conduct simple MD simulations to clarify the consistencies and linkages between our findings and those reported in the literatures. Since Raman band of cysteines 2 and 7 of hIAPP implies these species in primary monomeric or low-order oligomeric forms, we perform simulations on monomers to obtain atomic-level molecular structures using our limited computing power and point out the structural features of the low-order oligomers developed from the similar monomers in suggested references. Below is the revised main text with proper citations:

Newly added discussion with reference to the previous studies on these beta-sheet-containing hIAPP species in the manuscript:

In particular, we identified two types of low-populated transient species at the early stage of hIAPP aggregation under pH 7.4, which are consistent with the β -sheet containing oligomers observed in previous SERS, 2D-IR, and MD simulation studies^{57,58,62}. Specifically, the type I species possesses a turn structure (residues 20-25), which is counted as a critical loop-forming region to promote aggregation⁶². The type II species contains a short β -hairpin (residues 14-19 and 26-29), in accordance with the proposed amyloidogenic precursors and building blocks of amyloid fibrils in the literatures^{57,58}. Furthermore, it is reported that significant changes of conformational populations occurred in the early stage as the development from monomers to oligomers^{57,58}, followed by the increase of oligomeric structures persisting the complementary of turns and β -sheets in a partially ordered-disordered arrangement⁵⁸ to form amyloid fibrils. Hence, the direct characterizations of these early-stage transient species are crucial to resolve the amyloid aggregation of hIAPP.

Revised spectral analysis in connection with MD simulations on these beta-sheet-containing hIAPP species in the manuscript:

As evident from these spectral features, the type I spectra might be putatively assigned to the transient species of hIAPP containing the turn structure with an extended C-terminal, while the type II spectra might be assigned to the transient species containing both helix-coil and β -sheet structures with a constrained C-terminal. Notably, the spectra of type I and type II species both contain the peak at 523 cm^{-1} , indicative of the intact disulfide bond to stabilize the N-terminal helix structure. There is no obvious peak at 490 cm^{-1} , which is related to disulfide bond restrictions (Supplementary Fig. 15), implying these hIAPP species in primary monomeric or low-order oligomeric forms without forming compact aggregates or amyloid fibrils⁵². MD simulations of the monomeric hIAPP at pH 7.4 demonstrate the emerging turn and β -sheet fractions among the predominating helix and coil structures in the secondary structure evolution and content in Fig. 5i and 5k. Top three most-populated conformations are

illustrated in Fig. 5j. The predominant conformation exhibits the partial helix-coil structure. The type I minor conformation has a turn at residues 20-25. The type II minor conformation contains a short β -hairpin structure (residue 14-19 and 26-29), giving lower values of the root-mean-square fluctuation (RMSF) at residues 30-37 than that of the type I conformer to reflect a more constrained C-terminal (Supplementary Fig. 16). These results are in good agreement with the experimental observations and consistent with previous MD simulations⁵⁷⁻⁵⁹, providing molecular details of the monomeric species that might contribute to the type I and II spectra of hIAPP at the very early aggregation stage. In addition, since His-18 ($pK_{a3} = 6.0$) deprotonated at pH 7.4, it reduces electrostatic repulsion to promote intra- and inter-molecular interactions⁵⁴. As indicated by the distinct change of Tyr doublets in the spectra of type II species (red), Tyr-37 might interact with His-18 by hydrogen bonding and/or ring stacking to constrain C-terminal, which could further stabilize the β -sheet structure in type II transient species and drive the subsequent aggregation^{42,60}. This assignment is supported by the previous MD simulations that the interaction between His-18 and Tyr-37 would minimize the entropic cost of initial oligomerization⁶¹ and facilitate the intermolecular contact⁵⁸ upon aromatic stacking and hydrophobic collapse^{57,62} to assemble β -sheet-rich structures in low-order oligomers^{57,58,61,62}.

Revised Fig. 5 showing spectral analysis in connection with MD simulations:

Figure 5. ...c Representative SERS spectra of 10 μM hIAPP solution under pH 7.4 incubation

at $t = 0$ h (black) and $t = 2$ h (most-populated spectrum, green; type I rare spectrum, purple; type II rare spectrum, red). **d** The disulfide bond region of (c). **e** The Tyr doublets region of (c). **f** The Amide I region of (c). **g** Secondary derivative spectra of (f)...**i** Representative time evolution of secondary structure per residue of monomeric hIAPP with +3 charges at pH 7.4. Color code: blue: helix; white: random coil; green: bend; yellow: turn; red: β -sheet. **j** Snapshots of the predominant conformation, the type I minor conformation, and the type II minor conformation from (i). **k** Secondary structural contents of disordered, helix, turn, and β -sheet of the monomeric hIAPP from (i).

In addition, the authors performed multiple SERS measurements on the same or several independent samples. This is very important to establish, mainly if single-molecule experimental techniques are used to have a statistically significant sampling.

We appreciate the reviewer for highlighting one of the most valuable advantages of single-molecule experiments, which could enable the statistical analysis of individual molecules to discriminate sub-population fractions and reconstruct heterogeneous ensembles. Leveraging the high efficiency of our platform, we repeat the SERS measurements on several independent batches of hIAPP samples and acquire a statistically significant amount of SERS spectra during their incubations at different pH conditions (10000 spectra at each timepoint) to provide both ensemble structures and individual conformations with the unique information on probability, population, and distribution. As suggested, we ensure the statistical refinement of numerous repeated single-molecule SERS experiments on several independent samples to enhance the reliability and reproducibility for characterizing the low-populated species in complex mixtures.

Revised main text and method section on protein preparation and SERS characterization:

Fig.2a displays the spectral mapping out of 3600 SERS measurements from twenty parallel experimental sessions collected at the dynamic nanocavity in time series, showing the distinct spectral features in the regions of $590\text{-}610\text{ cm}^{-1}$ and $1620\text{-}1660\text{ cm}^{-1}$. Statistical analysis of the data indicates that $\sim 5\%$ spectra exhibited peaks intensities above three times the standard deviation of noise (Supplementary Fig. 7) and classified into the catalog of 'single-MB event', 'single-NBA event', or 'dual-MB and NBA event', based on the characteristic Raman peaks of MB or NBA at a high concentration of 10^{-5} M in Supplementary Fig. 8.

To reflect ensemble structures, 10000 SERS spectra were collected at each early timepoint during the incubations of hIAPP from different batches of independent preparations in multiple parallel experimental sessions.....To better analyze protein secondary structures, we conducted the second derivative analysis⁵² for the amide I region from 1550 cm^{-1} to 1750 cm^{-1} in Fig. 4f to plot Fig. 4g, and presented the mapping of these secondary derivative spectra in Supplementary Fig. 13 and Fig. 4h.....However, after 2-hour incubation at pH 7.4, we observed emerging sub-populations of

SERS spectra of hIAPP from a statistically significant sampling based on the efficient measurements at the dynamic nanocavity with 1 s accumulation time in multiple parallel experimental sessions, as indicated by the spectral mapping in Fig. 5h.

Based on the statistically significant sampling on our efficient single-molecule SERS platform in combined with MD simulations, two types of low-populated hIAPP transient species were differentiated from its predominant monomers at the early stage of pH-induced amyloid aggregation, which adopted the critical turn structure or the partial β -sheet with constrained C-terminal.

hIAPP preparation

In parallel experimental sessions, several batches of the hIAPP stock solutions were prepared as independent samples for the subsequent incubations at different pH conditions. Meanwhile, the time-dependent CD, SERS, and AFM characterizations were conducted repeatedly for parallel experimental sessions during the incubations of these hIAPP solutions from different batches of independent preparations.

Instrumental setup and dynamic SERS measurements

To characterize the structures of hIAPP, 10000 SERS spectra were collected at each early timepoint during the incubations under different pH conditions in multiple parallel experimental sessions for statistical analysis.

Revised Figures 2, 4, 5 on the spectral mapping from multiple parallel experimental sessions:

Figure 4. h Mapping of secondary derivative spectra of the Amide I region of the SERS spectra of hIAPP at $t = 2$ h incubation under pH 5.5 from parallel experimental sessions. The color bar shows the normalized intensities from low (dark blue) to high (red-black).

Figure 5. h Mapping of secondary derivative spectra of the Amide I region of the SERS spectra of hIAPP at $t = 2$ h incubation under pH 7.4 from parallel experimental sessions. The color bar shows the normalized intensities from low (dark blue) to high (red-black).

Figure 2. a Spectral mapping of 10^{-8} M MB and NBA mixture solution from 3600 SERS measurements across twenty parallel sessions. Integration time: 1 s per spectrum. Only the spectra with the peak intensities above three times the standard deviation of the noise are shown.

Revised Supplementary Fig. 13 and 14 on the spectral mapping from multiple parallel experimental sessions:

Supplementary Fig. 13. Mapping of secondary derivative spectra of the Amide I region of the SERS spectra of hIAPP at $t = 0$ h incubation under pH 5.5 from parallel experimental sessions. The color bar shows the normalized intensities from low (dark blue) to high (red-black).

Supplementary Fig. 14. Mapping of secondary derivative spectra of the Amide I region of the SERS spectra of hIAPP at $t = 0$ h incubation under pH 7.4 from parallel experimental sessions. The color bar shows the normalized intensities from low (dark blue) to high (red-black).

Reviewer #1 (Remarks to the Author):

The revisited version of the manuscript reports a significant effort by the authors to improve the quality and the significance according to the referees' comments. I am rather satisfied by the authors' reply, but some points are still not fully convincing.

1. As in my previous comment, figure 3 illustrates the comparison between experimental and DFT calculated spectra. The difference was significant and not well discussed in the manuscript, now the difference looks much lower and the agreement between the experiments and the model is good. Anyway, it is not fully clear if this due to an "arbitrary" correction of the data or to a more correct way to process the data. it is not clear why one should trust more to these new results with respect to the previous one

2.fig. 4i and 5i are very important and an important plus for the manuscript, anyway, as they are presented in the text they are not clear to a generic reader. The figure is confused and maybe too small. Another way to present the data could be tried. Moreover, MD simulations should procure a clear 3D view of the molecular structures, and I expected to see some examples in the paper

3. in the method section, hIAPP preparation, the authors say "Then, the purified hIAPP was rehydrated in 10 mM PBS buffer and filtered through a 0.22 nm Tuffryn syringe filter..."
0.22nm???

moreover it is not fully clear how they modified the pH of the solution.

I think that the manuscript reports some interesting an new results, but the overall quality still new some improvements before to be accepted in Nat. Comm.

Reviewer #2 (Remarks to the Author):

The authors clearly and comprehensively answered my queries. Now the manuscript sounds good.

We thank the reviewers for their thoughtful insights and valuable recommendations, and we have made the following supplementary revisions in response to their constructive comments:

Reviewer #1 (Remarks to the Author):

The revisited version of the manuscript reports a significant effort by the authors to improve the quality and the significance according to the referees' comments. I am rather satisfied by the authors' reply, but some points are still not fully convincing.

We appreciate the reviewer for recognizing the improvements and significance of our work.

1. As in my previous comment, figure 3 illustrates the comparison between experimental and DFT calculated spectra. The difference was significant and not well discussed in the manuscript, now the difference looks much lower and the agreement between the experiments and the model is good. Anyway, it is not fully clear if this due to an "arbitrary" correction of the data or to a more correct way to process the data. it is not clear why one should trust more to these new results with respect to the previous one.

DFT is extensively employed as a reliable and cost-effective quantum chemical method for the determination of vibrational frequencies. In particular, B-based DFT procedures with the 6-31G(d,p) basis set usually yield calculated wavenumbers close to their experimental values with scale factors approaching unity. These computed values can be directly used if minor discrepancies are tolerable. To reach high accuracy, B3-based DFT procedures with larger basis sets require supplementary corrections to compensate the inherent deficiencies listed below.

Based on the simple harmonic oscillator model, DFT computations of vibrational frequencies neglect anharmonicity in the vibrational potential energy surface and rely on approximations to solve the Schrödinger equation. Moreover, errors arise due to the systematic electronic structure insufficiencies, especially the incomplete description of electron correlation from the use of finite basis sets. Consequently, calculated frequencies are typically overestimated, which become more prominent at high wavenumbers. Taken these factors into consideration, scaling procedures utilize modest computational cost to improve accuracy, which are comparable to anharmonic calculations. Hence, empirical parameters (i.e. direct scaling by least-squares fits) are widely adopted to correct calculated wavenumbers for the interpretation of experimental vibrational spectra. (*J. Phys. Chem.* 100, 16502–16513 (1996); *Phys. Sci. Rev.* 3, 1–30 (2019))

Using a standard least-squares approach, we derive the scaling factors to correct calculated frequencies to experimental fundamentals in the wavenumber region above 1000 cm⁻¹, as

recommended in the literature. (*J. Comput. Chem.* 33, 2380–2387 (2012); *Appl. Spectrosc.* 63, 733–741 (2009); *J. Phys. Chem.* 99, 3093–3100 (1995)) The optimum scaling factor $\lambda=0.9770$ significantly improves the agreement between DFT computations and experiments, giving the relative deviations less than 1%. In particular, the tyrosine ν_{8a} mode (pH 1.0) is observed at 1620 cm^{-1} and calculated at 1635 cm^{-1} . The tyrosine ν_{8a} mode (pH 13.0) is observed at 1602 cm^{-1} and calculated at 1592 cm^{-1} . More importantly, the identical peak shifting trends between the DFT calculated and the experimental spectra provide a reliable support to the structural changes of tyrosine from +1 charged state to -2 charged state at different pH environments.

To enhance the rigor of this work, we update Supplementary Information to include the least-squares procedures for the corrections of the calculated and the experimental wavenumbers of Tyr in the high wavenumber region ($1000\text{-}2000\text{ cm}^{-1}$) at pH 1.0 and pH 13.0, respectively. We revise Fig. 3 and main text to clarify the correction procedures for improving the agreement between computations and experiments, which strengthens the assignment and interpretation of the vibrational modes of Tyr associated with its structural changes at different pH conditions.

Newly added supplementary information to elaborate the scaling procedures for the correction of the DFT calculated frequencies:

Supplementary Note 3. Least-squares procedures for wavenumber corrections

Based on a least-squares procedure^{7,8}, the optimum scaling factor λ that correlates calculated frequencies to experimental wavenumbers is determined by minimizing the residual Δ :

$$\Delta = \sum_i^N (\lambda \omega_i^{cal} - \tilde{\omega}_i^{exp})^2$$

Where λ is the scaling factor. N is the total number of vibrational modes in a designated wavenumber region. ω_i^{cal} and $\tilde{\omega}_i^{exp}$ are the i th calculated frequencies and experimental wavenumbers, respectively.

By substituting the calculated frequencies and the experimental wavenumbers of Tyr at +1 charged state under pH 1.0 condition in the high wavenumber region ($1000\text{-}2000\text{ cm}^{-1}$) into the equation, we can get:

$$\begin{aligned} \Delta_1 = & (1674\lambda_1 - 1620)^2 + (1604\lambda_1 - 1572)^2 + (1508\lambda_1 - 1434)^2 + (1380\lambda_1 - 1352)^2 \\ & + (1350\lambda_1 - 1333)^2 + (1308\lambda_1 - 1271)^2 + (1236\lambda_1 - 1217)^2 \\ & + (1150\lambda_1 - 1153)^2 + (1082\lambda_1 - 1073)^2 \end{aligned}$$

This function is quadratic in nature and its graphical representation forms a parabolic curve, as shown in Supplementary Fig. 12:

Supplementary Fig. 12. Residues as a function of scaling factors correlating the calculated frequencies and the experimental wavenumbers of Tyr at +1 charged state under pH 1.0 condition in the high wavenumber region (1000-2000 cm^{-1}).

Δ_1 reaches its minimum at the vertex of the quadratic function, denoted as $\frac{d\Delta_1}{d\lambda_1} = 0$, leading to:

$$\lambda_{1 \text{ opt}} = \frac{\sum_i^N (\omega_i^{\text{cal}} \tilde{\omega}_i^{\text{exp}})}{\sum_i^N (\omega_i^{\text{cal}})^2}$$

$$= \frac{1674 \times 1620 + 1604 \times 1572 + 1508 \times 1434 + 1380 \times 1352 + 1350 \times 1333 + 1308 \times 1271 + 1236 \times 1217 + 1150 \times 1153 + 1082 \times 1073}{1674^2 + 1604^2 + 1508^2 + 1380^2 + 1350^2 + 1308^2 + 1236^2 + 1150^2 + 1082^2}$$

$$= 0.9770$$

Therefore, the optimum scaling factor $\lambda_{1 \text{ opt}}$ that correlates the calculated frequencies and the experimental wavenumbers of Tyr at +1 charged state under pH 1.0 condition is 0.9770.

Similarly, by substituting the calculated frequencies and the experimental wavenumbers of Tyr at -2 charged state under pH 13.0 condition in the high wavenumber region (1000-2000 cm^{-1}) into the equation, we can get:

$$\Delta_2 = (1630\lambda_2 - 1602)^2 + (1594\lambda_2 - 1570)^2 + (1526\lambda_2 - 1442)^2 + (1480\lambda_2 - 1423)^2 + (1352\lambda_2 - 1338)^2 + (1290\lambda_2 - 1269)^2 + (1222\lambda_2 - 1216)^2 + (1182\lambda_2 - 1161)^2 + (1092\lambda_2 - 1071)^2$$

This quadratic function gives a parabolic graph as shown in Supplementary Fig. 13:

Supplementary Fig. 13. Residues as a function of scaling factors correlating the calculated frequencies and the experimental wavenumbers of Tyr at -2 charged state under pH 13.0 condition in the high wavenumber region (1000-2000 cm^{-1}).

Δ_2 reaches its minimum at the vertex of the quadratic function, denoted as $\frac{d\Delta_2}{d\lambda_2} = 0$, leading to:

$$\lambda_{2 \text{ opt}} = \frac{\sum_i^N (\omega_i^{\text{cal}} \tilde{\omega}_i^{\text{exp}})}{\sum_i^N (\omega_i^{\text{cal}})^2}$$

$$\begin{aligned} &= \frac{1630 \times 1602 + 1594 \times 1570 + 1526 \times 1442 + 1480 \times 1423 + 1352 \times 1338 + 1290 \times 1269 + 1222 \times 1216 + 1182 \times 1161 + 1092 \times 1071}{1630^2 + 1594^2 + 1526^2 + 1480^2 + 1352^2 + 1290^2 + 1222^2 + 1182^2 + 1092^2} \\ &= 0.9770 \end{aligned}$$

Therefore, the optimum scaling factor $\lambda_{2 \text{ opt}}$ that correlates the calculated frequencies and the experimental wavenumbers of Tyr at -2 charged state under pH 13.0 condition is 0.9770.

Revised main text to clarify the origin and the correction procedure for the discrepancy between calculated and experimental wavenumbers:

Due to the inherent deficiencies related to DFT simulations (i.e. neglect of anharmonicity, incomplete incorporation of electron correlation, and other approximations), the calculated frequencies were overestimated especially in higher wavenumber region^{46,47}. Based on the least-squares procedure^{48,49} (Supplementary Note. 3), an empirical scaling factor 0.9770 was applied to the wavenumbers above 1000 cm⁻¹ in the simulated Raman spectra of Tyr at different charge states to align with the experimental observations at pH 1.0 and pH 13.0, respectively.

New Fig 3c comparing the experimental spectra and the DFT calculated spectra of tyrosine:

Figure 3. pH-dependent structural transitions of Tyr. ... c Comparison of SERS spectra (upper panel) and simulated Raman spectra (lower panel) of Tyr at pH 1.0 (black) and pH 13.0 (red), respectively. The SERS spectra are adopted from (a). The simulated Raman spectra and electrostatic potential are generated from DFT calculations on Tyr in +1 charged state as NH₃⁺CH(CH₂C₆H₄OH)COOH (d) and -2 charged state as NH₂CH(CH₂C₆H₄O⁻)COO⁻ (e), respectively. Based on least-squares procedures, a scaling factor of 0.9770 was applied to the wavenumbers above 1000 cm⁻¹ in simulated Raman spectra.

2.fig. 4i and 5i are very important and an important plus for the manuscript, anyway, as they are presented in the text they are not clear to a generic reader. The figure is confused and maybe too small. Another way to present the data could be tried. Moreover, MD simulations should provide a clear 3D view of the molecular structures, and I expected to see some examples in the paper

Following the reviewer's suggestion, we revise Fig. 4i and 5i to annotate different secondary

structures with color legends in addition to text legends for a clearer presentation of hIAPP's secondary structure evolutions over time, which align with the colors in the statistical analysis of its secondary structural contents in Fig. 4k and 5k and the snapshots of its representative conformations in Fig. 4j and 5j. To improve clarity, we enlarge the dimensions of Fig. 4i and 5i while adding the amino acid abbreviations labeled in Fig. 4a and 5a along the residue index in Y-axis to depict how the secondary structures of hIAPP change over time at residue level. For better illustration, we incorporate the snapshots of the representative conformations with their corresponding time frames in Fig. 4j and 5j adjacent to the secondary structure evolutions of hIAPP in Fig. 4i and 5i. Moreover, we offer the vivid representations of the 3D structural dynamics of hIAPP in the form of two supplementary short videos, generated by animating the trajectory frames from Fig. 4i and 5i, respectively. Furthermore, to facilitate a comprehensive visualization of the representative conformations of hIAPP in Fig. 4j and 5j, including initial conformations, predominant conformations, minor conformations, and final conformations, we show the snapshots from different 3D coordinate perspectives together with the short videos demonstrating the 360-degree rotations of these conformations in Supplementary Information. Besides, we provide the PDB files encompassing all conformations of hIAPP extracted from MD simulation trajectories to support visual reproduction and post-processing of them using a wide range of visualization software tools for potential follow-up studies in the future.

Revised Fig. 4a, 4i, 4j, and 4k for a more informative presentation of MD simulation results incorporating the properties of hIAPP residues with +4 charges at pH 5.5, the time evolution of secondary structure per residue, the representative conformations with their corresponding time frames along the trajectory, and the statistical analysis of secondary structural contents:

Figure 4. Structural characterizations of hIAPP incubated under pH 5.5. a Properties of hIAPP residues with +4 charges at pH 5.5: Positively charged residues in red, hydrophobic

residues in blue, and residues with helical potency in the dashed box....i Representative time evolution of secondary structure per residue of monomeric hIAPP with +4 charges at pH 5.5. Color legends for secondary structures: White: Coil; Green: Bend; Blue: α -Helix; Grey: 3-Helix; Yellow: Turn; Red: β -Sheet; Black: β -Bridge. j Snapshots of the initial conformation (0 ns), predominant conformation (68 ns), and final conformation (200 ns) from (i). k Statistical analysis of the secondary structural contents of the monomeric hIAPP from (i).

Newly added supplementary short video to show the structural dynamics of monomeric hIAPP with +4 charges at pH 5.5 by animating the trajectory frames from Fig. 4i:

The file titled “Supplementary Movie 1.mov” is attached in Supplementary Information. To provide a preview of the video content, representative frames are shown below.

The PDB file titled “Supplementary Data 1.pdb” is attached in Supplementary Information. The PDB file encompasses all conformations extracted from the MD simulation trajectory of monomeric hIAPP with +4 charges at pH 5.5 in Fig. 4i.

Newly added supplementary figures showing the snapshot of representative conformations of hIAPP in Fig. 4j from different 3D coordinate perspectives and short videos demonstrating the

360-degree rotations of these conformations:

Supplementary Fig. 24. Snapshots from x-y, y-z, and x-z plane perspectives of the initial conformation, the predominant conformation and the final conformation in the representative simulation trajectory of monomeric hIAPP with +4 charges at pH 5.5 shown in Fig. 4j.

The file titled “Supplementary Movie 2.mov” is attached in Supplementary Information. To provide a preview of the video content, representative frames are shown below.

The file titled “Supplementary Movie 3.mov” is attached in Supplementary Information. To provide a preview of the video content, representative frames are shown below.

The file titled “Supplementary Movie 4.mov” is attached in Supplementary Information. To provide a preview of the video content, representative frames are shown below.

Revised Fig. 5a, 5i, 5j, and 5k for a more informative presentation of MD simulation results incorporating the properties of hIAPP residues with +3 charges at pH 7.4, the time evolution of secondary structure per residue, the representative conformations with their corresponding time frames along the trajectory, and the statistical analysis of secondary structural contents:

Figure 5. Structural characterizations of hIAPP incubated under pH 7.4. a Properties of hIAPP residues with +3 charges at pH 7.4: Positively charged residues in red, hydrophobic residues in blue, and His-18 highlighted in yellow as the deprotonated residue upon the change of pH from 5.5 to 7.4...**i** Representative time evolution of secondary structure per residue of monomeric hIAPP with +3 charges at pH 7.4. Color legends for secondary structures: White: Coil; Green: Bend; Blue: α -Helix; Grey: 3-Helix; Yellow: Turn; Red: β -Sheet; Black: β -Bridge. **j** Snapshots of the initial conformation (0 ns), type II minor conformation (49 ns), predominant conformation (112 ns), type I minor conformation (171 ns) and final conformation (200 ns) from (i). **k** Statistical analysis of the secondary structural contents of the monomeric hIAPP from (i).

Newly added supplementary short video to show the structural dynamics of monomeric hIAPP with +3 charges at pH 7.4 by animating the trajectory frames from Fig. 5i:

The file titled “Supplementary Movie 5.mov” is attached in Supplementary Information. To provide a preview of the video content, representative frames are shown below.

The PDB file titled “Supplementary Data 2.pdb” is attached in Supplementary Information. The PDB file encompasses all conformations extracted from the MD simulation trajectory of monomeric hIAPP with +3 charges at pH 7.4 in Fig. 5i.

Newly added supplementary figures showing the snapshot of representative conformations of hIAPP in Fig. 5j from different 3D coordinate perspectives and short videos demonstrating the 360-degree rotations of these conformations:

Supplementary Fig. 25. Snapshots from x-y, y-z, and x-z plane perspectives of the initial conformation, the type II minor conformation, the predominant conformation, the type I minor conformation, and the final conformation in the representative simulation trajectory of monomeric hIAPP with +3 charges at pH 7.4 shown in Fig. 5j.

The file titled “Supplementary Movie 6.mov” is attached in Supplementary Information. To provide a preview of the video content, representative frames are shown below.

The file titled “Supplementary Movie 7.mov” is attached in Supplementary Information. To provide a preview of the video content, representative frames are shown below.

The file titled “Supplementary Movie 8.mov” is attached in Supplementary Information. To provide a preview of the video content, representative frames are shown below.

The file titled “Supplementary Movie 9.mov” is attached in Supplementary Information. To provide a preview of the video content, representative frames are shown below.

The file titled “Supplementary Movie 10.mov” is attached in Supplementary Information. To provide a preview of the video content, representative frames are shown below.

Revised Methods section incorporating the representations of the 3D structures of hIAPP:

Molecular Dynamic (MD) simulation

... In addition to the secondary structure evolutions of hIAPP shown in Fig. 4i and 5i, two supplementary short videos are generated by animating the trajectory frames from MD simulations. The representative conformations of hIAPP along the simulation trajectories

shown in Fig. 4j and 5j, including initial conformations, predominant conformations, minor conformations, and final conformations, are further illustrated from x-y, y-z, and x-z plane perspectives in Supplementary Fig. 24 and 25, together with the short videos demonstrating the 360-degree rotations of these conformations in Supplementary Information. The PDB files encompassing all conformations of hIAPP extracted from MD simulation trajectories are provided in Supplementary Information.

3. in the method section, hIAPP preparation, the authors say "Then, the purified hIAPP was rehydrated in 10 mM PBS buffer and filtered through a 0.22 nm Tuffryn syringe filter..." 0.22nm??? moreover it is not fully clear how they modified the pH of the solution.

We thank the reviewer for catching the incorrect unit, "0.22 nm", referring to the pore size of the Tuffryn syringe filter. We have rectified this error and specified the correct pore size of the syringe filter as "0.22 μm " in the revised manuscript. In addition, we thoroughly review all the units in the manuscript to ensure accurate reporting of quantitative details.

Moreover, we include the details on the preparation and adjustment of hIAPP solutions at the desired pH in the Methods section. Initially, hIAPP stock solutions were diluted using the PBS buffers with the desired pH at 5.5 or 7.4, which were fine-tuned using 1 M HCl or 1 M NaOH (if necessary) under the monitoring of a pH meter. During the two-phase incubation, the pH of the solution was adjusted from pH 7.4 to 5.5 by gradually adding drops of 1 M HCl under the continuous monitoring of a pH meter to ensure accurate pH control. According to calculations and observations, approximately 40 μL of 1 M HCl was sufficient to adjust the pH of a 10 mL hIAPP solution from 7.4 to 5.5, which could minimize the change of solution volume.

Revised Methods section to correct the unit and clarify the details of the pH adjustment:

hIAPP preparation

...Then, the purified hIAPP was rehydrated in PBS buffer (10 mM phosphate, 150 mM NaCl, pH 7.4) and filtered through a 0.22 μm Tuffryn syringe filter to prepare the stock solutions with a concentration of 1 mg/mL (250 μM). In parallel experimental sessions, several batches of the hIAPP stock solutions were prepared as independent samples for the subsequent incubations at different pH conditions. Portions of the hIAPP stock solutions were diluted with PBS buffer (10 mM phosphate, 150 mM NaCl, pH 5.5) to achieve a concentration of 10 μM and adjusted to pH 5.5 with 1 M HCl (if necessary) under the monitoring of a pH meter, followed by the 24-hour incubation at 37 °C. Portions of the hIAPP stock solutions were diluted with PBS buffer (10 mM phosphate, 150 mM NaCl, pH 7.4) to achieve a concentration of 10 μM and adjusted to pH 7.4 with 1 M HCl/1 M NaOH (if necessary) under the monitoring of a pH meter, followed

by the 24-hour incubation at 37 °C. Portions of the hIAPP stock solutions were diluted with PBS buffer (10 mM phosphate, 150 mM NaCl, pH 7.4) to achieve a concentration of 10 μM and adjusted to pH 7.4 with 1 M HCl/1 M NaOH (if necessary) under the monitoring of a pH meter, followed by the 2-hour incubation at 37 °C. Subsequently, the pH of this solution was further adjusted from pH 7.4 to 5.5 through the dropwise addition of 1 M HCl under the monitoring of a pH meter, followed by the succeeding 22-hour incubation at 37 °C...

I think that the manuscript reports some interesting an new results, but the overall quality still new some improvements before to be accepted in Nat. Comm.

We are grateful for the reviewer's comments affirming the interest and novelty of our study. Guided by the constructive feedback, we have thoroughly revised the manuscript to enhance clarity and ensure accuracy, which are the critical aspects for improving the overall quality of the manuscript and aligning it with the rigorous standards in Nat. Comm.

Reviewer #2 (Remarks to the Author):

The authors clearly and comprehensively answered my queries. Now the manuscript sounds good.

We would like to express our gratitude for the reviewer's thorough evaluation of our work.

Reviewer #1 (Remarks to the Author):

The authors did a great job to improve the manuscript according to my comments.

The manuscript can be now accepted

Reviewer #1 (Remarks to the Author):

The authors did a great job to improve the manuscript according to my comments.

The manuscript can be now accepted

We sincerely appreciate the reviewer's diligent assessments and constructive suggestions for enhancing the quality and impact of our study.